# Improving Diffusion Models for Inverse Problems using Manifold Constraints

**Hyungjin Chung**[*,1]    **Byeongsu Sim**[*,2]    **Dohoon Ryu**[1]    **Jong Chul Ye**[3,1,2]

[1] Dept. of Bio and Brain Engineering
[2] Dept. of Mathematical Sciences
[3]Kim Jaechul Graduate School of AI
[*]Equal contribution
Korea Advanced Institute of Science and Technology (KAIST)
{hj.chung, byeongsu.s, dh.ryu, jong.ye}@kaist.ac.kr

## Abstract

Recently, diffusion models have been used to solve various inverse problems in an unsupervised manner with appropriate modifications to the sampling process. However, the current solvers, which recursively apply a reverse diffusion step followed by a projection-based measurement consistency step, often produce sub-optimal results. By studying the generative sampling path, here we show that current solvers throw the sample path off the data manifold, and hence the error accumulates. To address this, we propose an additional correction term inspired by the manifold constraint, which can be used synergistically with the previous solvers to make the iterations close to the manifold. The proposed manifold constraint is straightforward to implement within a few lines of code, yet boosts the performance by a surprisingly large margin. With extensive experiments, we show that our method is superior to the previous methods both theoretically and empirically, producing promising results in many applications such as image inpainting, colorization, and sparse-view computed tomography. Code available here

## 1  Introduction

Diffusion models have shown impressive performance both as generative models themselves [41, 13], and also as unsupervised inverse problem solvers [41, 8, 9, 25] that do not require problem-specific training. Specifically, given a pre-trained unconditional score function (i.e. denoiser), solving the reverse stochastic differential equation (SDE) numerically would amount to sampling from the data generating distribution [41]. For many different inverse problems (e.g. super-resolution [8, 9], inpainting [41, 9], compressed-sensing MRI (CS-MRI) [40, 9], sparse view CT (SV-CT) [40], etc.), it was shown that simple incorporation of the measurement process produces satisfactory conditional samples, even when the model was not trained for the specific problem.

Nevertheless, for certain problems (e.g. inpainting), currently used algorithms often produce unsatisfactory results when implemented naively (e.g. boundary artifacts, as shown in Fig. 1 (b)). The authors in [32] showed that in order to produce high quality reconstructions, one needs to iterate back and forth between the noising and the denoising step at least $> 10$ times *per iteration*. These iterations are computationally demanding and should be avoided, considering that diffusion models are slow to sample from even without such iterations. On the other hand, a classic result of Tweedie's formula [37, 42] shows that one can perform Bayes optimal denoising in one step, once we know the gradient of the log density. Extending such result, it was recently shown that one can indeed

36th Conference on Neural Information Processing Systems (NeurIPS 2022).

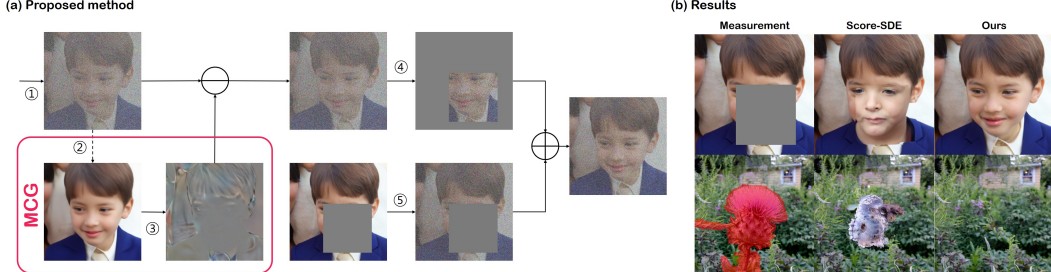

**(a) Proposed method**

**(b) Results**

Figure 1: Visual schematic of the MCG correction step. (a) ① Unconditional reverse diffusion generates $\boldsymbol{x}_i$; ② $Q_i$ maps the noisy $\boldsymbol{x}_i$ to generate $\hat{\boldsymbol{x}}_0$; ③ Manifold Constrained Gradient (MCG) $\frac{\partial}{\partial \boldsymbol{x}_i} \|\boldsymbol{W}(\boldsymbol{y} - \boldsymbol{H}\hat{\boldsymbol{x}}_0)\|_2^2$ is applied to fix the iteration on manifold; ④ Takes the orthogonal complement; ⑤ Samples from $p(\boldsymbol{y}_i|\boldsymbol{y})$, then combines $\boldsymbol{A}\boldsymbol{x}'_{i-1}$ and $\boldsymbol{y}_i$. (b) Representative results of inpainting, compared with score-SDE [41]. Reconstructions with score-SDE produce incoherent results, while our method produces high fidelity solutions.

perform a single-step denoising with learned score functions for denoising problems from the general exponential family [28].

In this work, we leverage the denoising result through Tweedie's formula and show that such denoised samples can be the key to significantly improving the performance of reconstruction using diffusion models across arbitrary linear inverse problems, despite the simplicity in the implementation. Moreover, we theoretically prove that if the score function estimation is globally optimal, the correction term from the manifold constraint enforces the sample path to stay on the plane tangent to the data manifold[1], so by combining with the reverse diffusion step, the solution becomes more stable and accurate.

## 2 Related Works

### 2.1 Diffusion Models

**Continuous Form**   For a continuous diffusion process $\boldsymbol{x}(t) \in \mathbb{R}^n$, $t \in [0, 1]$, we set $\boldsymbol{x}(0) \sim p_0(\boldsymbol{x}) = p_{data}$, where $p_{data}$ represents the data distribution of interest, and $\boldsymbol{x}(1) \sim p_1(\boldsymbol{x})$, with $p_1(\boldsymbol{x})$ approximating spherical Gaussian distribution, containing no information of data. Here, the forward noising process is defined with the following Itô stochastic differential equation (SDE) [41]:

$$d\boldsymbol{x} = \bar{\boldsymbol{f}}(\boldsymbol{x}, t)dt + \bar{g}(t)d\boldsymbol{w}, \tag{1}$$

with $\bar{\boldsymbol{f}} : \mathbb{R}^d \mapsto \mathbb{R}^d$ defining the linear drift function, $\bar{g}(t) : \mathbb{R} \mapsto \mathbb{R}$ defining a scalar diffusion coefficient, and $\boldsymbol{w} \in \mathbb{R}^n$ denoting the standard $n-$dimensional Wiener process. The forward SDE in (1) is coupled with the following reverse SDE by the Anderson's theorem [1, 41]:

$$d\boldsymbol{x} = [\bar{\boldsymbol{f}}(\boldsymbol{x}, t) - \bar{g}(t)^2 \nabla_{\boldsymbol{x}} \log p_t(\boldsymbol{x})]dt + \bar{g}(t)d\bar{\boldsymbol{w}}, \tag{2}$$

with $dt$ denoting the infinitesimal negative time step, and $\bar{\boldsymbol{w}}$ defining the standard Wiener process running backward in time. Note that the reverse SDE defines the generative process through the score function $\nabla_{\boldsymbol{x}} \log p_t(\boldsymbol{x})$, which in practice, is typically replaced with $\nabla_{\boldsymbol{x}} \log p_{0t}(\boldsymbol{x}(t)|\boldsymbol{x}(0))$ to minimize the following denoising score-matching objective

$$\min_{\theta} \mathbb{E}_{t \sim U(\varepsilon, 1), \boldsymbol{x}(0) \sim p_0(\boldsymbol{x}), \boldsymbol{x}(t) \sim p_{0t}(\boldsymbol{x}(t)|\boldsymbol{x}(0))} \left[ \|\boldsymbol{s}_{\theta}(\boldsymbol{x}(t), t) - \nabla_{\boldsymbol{x}_t} \log p_{0t}(\boldsymbol{x}(t)|\boldsymbol{x}(0))\|_2^2 \right]. \tag{3}$$

Once the parameter $\theta^*$ for the score function is estimated, one can replace the score function in (2) with $s_{\theta^*}(\boldsymbol{x}(t), t)$ to solve the reverse SDE [41].

**Discrete Form**   Due to the linearity of $\bar{\boldsymbol{f}}$ and $\bar{g}$, the forward diffusion step can be implemented with a simple reparameterization trick [29]. Namely, the general form of the forward diffusion is

$$\boldsymbol{x}_i = a_i \boldsymbol{x}_0 + b_i \boldsymbol{z}, \quad \boldsymbol{z} \sim \mathcal{N}(0, \boldsymbol{I}), \tag{4}$$

---

[1]We coin our method **M**anifold **C**onstrained **G**radient (MCG).

where we have replaced the continuous index $t \in [0, 1]$ with the discrete index $i \in \mathbb{N}$. On the other hand, the discrete reverse diffusion step can in general be represented as

$$\boldsymbol{x}_{i-1} = \boldsymbol{f}(\boldsymbol{x}_i, \boldsymbol{s}_{\theta^*}) + g(\boldsymbol{x}_i)\boldsymbol{z}, \quad \boldsymbol{z} \sim \mathcal{N}(0, \boldsymbol{I}), \tag{5}$$

where we have replaced the ground truth score function with the trained one. We detail the choice of $a_i, b_i, \boldsymbol{f}, g$ in Appendix. B.

## 2.2 Conditional Generative models for Inverse problems

The main problem of our interest in this paper is the inverse problem, retrieving the unknown $\boldsymbol{x} \in \mathbb{R}^n$ from a measurement $\boldsymbol{y}$:

$$\boldsymbol{y} = \boldsymbol{H}\boldsymbol{x} + \boldsymbol{\epsilon}, \quad \boldsymbol{y} \in \mathbb{R}^m, \boldsymbol{H} \in \mathbb{R}^{m \times n}, \tag{6}$$

where $\boldsymbol{\epsilon} \in \mathbb{R}^m$ is the noise in the measurement. Accordingly, for the case of the inverse problems, our goal is to generate samples from a conditional distribution with respect to the measurement $\boldsymbol{y}$, i.e. $p(\boldsymbol{x}|\boldsymbol{y})$. Accordingly, the score function $\nabla_{\boldsymbol{x}} \log p_t(\boldsymbol{x})$ in (2) should be replaced by the conditional score $\nabla_{\boldsymbol{x}} \log p_t(\boldsymbol{x}|\boldsymbol{y})$. Unfortunately, this strictly restricts the generalization capability of the neural network since the conditional score should be retrained whenever the conditions change. To address this, recent conditional diffusion models [22, 41, 8, 9] utilize the unconditional score function $\nabla_{\boldsymbol{x}} \log p_t(\boldsymbol{x})$ but rely on a projection-based measurement constraint to impose the conditions. Specifically, one can apply the following:

$$\boldsymbol{x}'_{i-1} = \boldsymbol{f}(\boldsymbol{x}_i, s_\theta) + g(\boldsymbol{x}_i)\boldsymbol{z}, \quad \boldsymbol{z} \sim \mathcal{N}(0, \boldsymbol{I}), \tag{7}$$
$$\boldsymbol{x}_{i-1} = \boldsymbol{A}\boldsymbol{x}'_{i-1} + \boldsymbol{b}_i, \tag{8}$$

where $\boldsymbol{A}, \boldsymbol{b}_i$ are functions of $\boldsymbol{H}, \boldsymbol{y}$, and $\boldsymbol{x}_0$. Note that (7) is identical to the unconditional reverse diffusion step in (5), whereas (8) effectively imposes the condition. It was shown in [9] that any general contraction mapping (e.g. projection onto convex sets, gradient step) may be utilized as (8) to impose the constraint.

Another recent work [25] advancing [26] establishes the state-of-the-art (SOTA) in solving *noisy* inverse problems with unconditional diffusion models, by running the conditional reverse diffusion process in the spectral domain achieved by performing singular value decomposition (SVD), and leveraging approximate gradient of the log likelihood term in the spectral space. The authors show that feasible solutions can be obtained with as small as 20 diffusion steps.

Prior to the development of diffusion models, Plug-and-Play (PnP) models [47, 53, 44] were used in a similar fashion by utilizing a general-purpose unconditional denoiser in the place of proximal mappings in model-based iterative reconstruction methods [5, 3]. Similarly, outside the context of diffusion models, iterative denoising followed by projection-based data consistency was proposed in [44]. In such view, diffusion models can be understood as generative variant of PnPs trained with multiple scales of noise.

GAN-based solvers are also widly explored [4, 10, 20], where the pre-trained generators are tuned at the test time by optimizing over the latent, the parameters, or jointly.

## 2.3 Tweedie's formula for denoising

In the case of Gaussian noise, a classic result of Tweedie's formula [37] tells us that one can achieve the denoised result by computing the posterior expectation:

$$\mathbb{E}[\boldsymbol{x}|\tilde{\boldsymbol{x}}] = \tilde{\boldsymbol{x}} + \sigma^2 \nabla_{\tilde{\boldsymbol{x}}} \log p(\tilde{\boldsymbol{x}}), \tag{9}$$

where the noise is modeled by $\tilde{\boldsymbol{x}} \sim \mathcal{N}(\boldsymbol{x}, \sigma^2 I)$. If we consider a diffusion model in which the forward step is modeled as $\boldsymbol{x}_i \sim \mathcal{N}(a_i\boldsymbol{x}_0, b_i^2 I)$ (discrete form), the Tweedie's formula can be rewritten as:

$$\mathbb{E}[\boldsymbol{x}_0|\boldsymbol{x}_i] = (\boldsymbol{x}_i + b_i^2 \nabla_{\boldsymbol{x}_i} \log p(\boldsymbol{x}_i))/a_i. \tag{10}$$

Tweedie's formula is in fact not only relevant to Gaussian denoising in the Bayesian framework, but have also been extended to be in close relation with kernel regression [34]. Moreover, it was shown that it can be applied to arbitrary exponential noise distributions beyond Gaussian [14, 28]. In the following, we use this key property to develop our algorithm.

# 3 Conditional Diffusion using Manifold Constraints

Although our original motivation of using the measurement constraint step in (8) was to utilize the unconditionally trained score function in the reverse diffusion step in (7), there is room for imposing additional constraints while still using the unconditionally trained score function.

Specifically, the Bayes rule $p(\boldsymbol{x}|\boldsymbol{y}) = p(\boldsymbol{y}|\boldsymbol{x})p(\boldsymbol{x})/p(\boldsymbol{y})$ leads to

$$\nabla_{\boldsymbol{x}} \log p(\boldsymbol{x}|\boldsymbol{y}) = \nabla_{\boldsymbol{x}} \log p(\boldsymbol{x}) + \nabla_{\boldsymbol{x}} \log p(\boldsymbol{y}|\boldsymbol{x}). \tag{11}$$

Hence, the score function in the reverse SDE in (7) can be replaced by (11), leading to

$$\boldsymbol{x}'_{i-1} = \boldsymbol{f}(\boldsymbol{x}_i, \boldsymbol{s}_\theta) - \alpha \frac{\partial}{\partial \boldsymbol{x}_i} \|\boldsymbol{W}(\boldsymbol{y} - \boldsymbol{H}\boldsymbol{x}_i)\|_2^2 + g(\boldsymbol{x}_i)\boldsymbol{z}, \quad \boldsymbol{z} \sim \mathcal{N}(0, \boldsymbol{I}) \tag{12}$$

where $\alpha$ and $\boldsymbol{W}$ depend on the noise covariance, if the noise $\boldsymbol{\epsilon}$ in (6) is Gaussian.

Now, one of the important contributions of this paper is to reveal that the Bayes optimal denoising step in (10) from the Tweedie's formula leads to a preferred condition both empirically and theoretically. Specifically, we define the set constraint for $\boldsymbol{x}_i$, called the *manifold constrained gradient (MCG)*, so that the gradient of the measurement term stays on the manifold (see Theorem 1):

$$\boldsymbol{x} \in \mathcal{X}_i, \quad \text{where} \quad \mathcal{X}_i = \{\boldsymbol{x} \in \mathbb{R}^n \mid \boldsymbol{x} = (\boldsymbol{x} + b_i^2 \boldsymbol{s}_\theta(\boldsymbol{x}, i))/a_i\} \tag{13}$$

To deal with the potential deviation from the measurement consistency, we again impose the data consistency step (8). Putting them together, the discrete reverse diffusion under the additional manifold constraint and the data consistency can be represented by

$$\boldsymbol{x}'_{i-1} = \boldsymbol{f}(\boldsymbol{x}_i, \boldsymbol{s}_\theta) - \alpha \frac{\partial}{\partial \boldsymbol{x}_i} \|\boldsymbol{W}(\boldsymbol{y} - \boldsymbol{H}\hat{\boldsymbol{x}}_0(\boldsymbol{x}_i))\|_2^2 + g(\boldsymbol{x}_i)\boldsymbol{z}, \quad \boldsymbol{z} \sim \mathcal{N}(0, \boldsymbol{I}), \tag{14}$$

$$\boldsymbol{x}_{i-1} = \boldsymbol{A}\boldsymbol{x}'_{i-1} + \boldsymbol{b}. \tag{15}$$

We illustrate our scheme visually in Fig. 1 (a), specifically for the task of image inpainting. The additional step leads to a dramatic performance boost, as can be seen in Fig. 1 (b). Note that while the mapping (10) does not rely on the measurement, our gradient term in (14) incorporates the information of $\boldsymbol{y}$ so that the gradient of the measurement terms stays on the manifold. In the following, we study the theoretical properties of the method. Further algorithmic details and adaptations to each problem that we tackle are presented in Section C.

We note that the authors of [19] proposed a similar gradient method for the application of temporal imputation and super-resolution. When combining (14) with (15), one can arrive at a similar gradient method proposed in [19], and hence our method can be seen as a generalization to arbitrary linear inverse problems. Furthermore, there are vast literature in the context of PnP models that utilize pre-trained denoisers together with gradient of the log-likelihood to solve inverse problems [30, 48, 11]. Among them, [30] is especially relevant to this work since their method relies on modified Langevin diffusion, together with Tweedie's denoising and projections to the measurement subspace.

# 4 Geometry of Diffusion Models and Manifold Constrained Gradient

In this section, we theoretically support the effectiveness of the proposed algorithm by showing the problematic behavior of the earlier algorithm and how the proposed algorithm resolves the problem. We defer all proofs in the supplementary section. To begin with, we borrow a geometrical viewpoint of the data manifold.

**Notation** For a scalar $a$, points $\boldsymbol{x}, \boldsymbol{y}$ and a set $A$, we use the following notations. $aA := \{a\boldsymbol{x} : \boldsymbol{x} \in A\}$; $d(\boldsymbol{x}, A) := \inf_{\boldsymbol{y} \in A} \|\boldsymbol{x} - \boldsymbol{y}\|_2$; $B_r(A) := \{\boldsymbol{x} : d(\boldsymbol{x}, A) < r\}$; $T_{\boldsymbol{x}}\mathcal{M}$: the tangent space to a manifold $\mathcal{M}$ at $\boldsymbol{x}$; $\boldsymbol{J}_f$: the Jacobian matrix of a vector valued function $f$. We define $p_0 = p_{data}$.

To develop the theory, we need an assumption on the data distribution.

**Assumption 1** (Strong manifold assumption: linear structure). *Suppose $\mathcal{M} \subset \mathbb{R}^n$ is the set of all data points, here we call the data manifold. Then, the manifold coincides with the tangent space with dimension $l \ll n$.*

$$\mathcal{M} \cap B_R(\boldsymbol{x}_0) = T_{x_0}\mathcal{M} \cap B_R(x_0) \text{ and } T_{x_0}\mathcal{M} \cong \mathbb{R}^l.$$

*Moreover, the data distribution $p_0$ is the uniform distribution on the data manifold $\mathcal{M}$.*

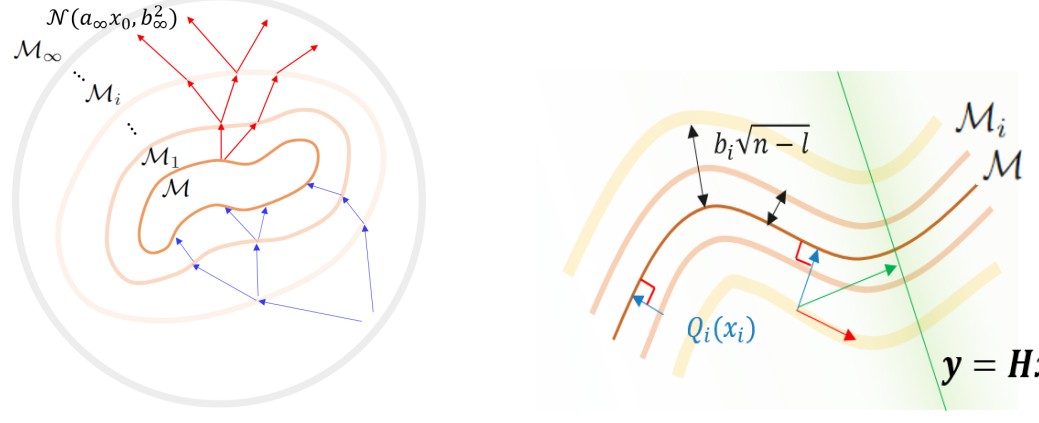

(a) Geometry of diffusion model          (b) MCG correction

Figure 2: In both (a) and (b), the central manifolds represent the data manifold $\mathcal{M}$, encircled by manifolds of noisy data $\mathcal{M}_i$. The concentration on the manifold of noisy data and the distance from the clean data manifold are prescribed by Proposition 1. In (a), the backward (resp. forward) step depicted by blue (resp. red) arrows can be considered as transitions from $\mathcal{M}_i$ to $\mathcal{M}_{i-1}$ (resp. $\mathcal{M}_{i-1}$ to $\mathcal{M}_i$). In (b), arrows refer to the directions of conventional projection onto convex sets (POCS) step (green arrow) and MCG step (red arrow) which can be predicted by Theorem 1.

We need to recall that the conventional manifold assumption is about the intrinsic geometry of data points having a low dimensional nature. However, we assume more in this work: the manifold is locally linear. Although this stronger assumption might narrow the practice of the theory, the geometric approach may provide new insights on diffusion models. Under this assumption, the following proposition shows how the data perturbed by noise lies in the ambient space, illustrated pictorially in Fig. 2a.

**Proposition 1** (Concentration of noisy data). *Consider the distribution of noisy data $p_i(\boldsymbol{x}_i) = \int p(\boldsymbol{x}_i|\boldsymbol{x})p_0(\boldsymbol{x})d\boldsymbol{x}, p(\boldsymbol{x}_i|\boldsymbol{x}) \sim \mathcal{N}(a_i\boldsymbol{x}, b_i^2\boldsymbol{I})$. Then $p_i(\boldsymbol{x}_i)$ is concentrated on $(n-1)$-dim manifold $\mathcal{M}_i := \{\boldsymbol{y} \in \mathbb{R}^n : d(\boldsymbol{y}, a_i\mathcal{M}) = r_i := b_i\sqrt{n-l}\}$. Rigorously, $p_i(B_{\epsilon r_i}(\mathcal{M}_i)) > 1 - \delta$, for some small $\epsilon, \delta > 0$.*

**Remark 1** (Geometric interpretation of the diffusion process). *Considering Proposition 1, the manifolds of noisy data can be interpreted as interpolating manifolds between the two: the hypersphere, where pure noise $\mathcal{N}(a_\infty\boldsymbol{x}_0, b_\infty^2)$ is concentrated, and the clean data manifold. In this regard, the diffusion steps are mere transitions from one manifold to another and the diffusion process is a transport from the data manifold to the hypersphere through interpolating manifolds. See Fig. 2a.*

**Remark 2.** *We can infer from the proposition that the score functions are trained only with the data points concentrated on the noisy data manifolds. Therefore, inaccurate inference might be caused by application of a score function on points away from the noisy data manifold.*

**Proposition 2** (score function). *Suppose $s_\theta$ is the minimizer of the denoising score matching loss in (3). Let $Q_i$ be the function that maps $\boldsymbol{x}_i$ to $\hat{\boldsymbol{x}}_0$ for each $i$,*

$$Q_i : \mathbb{R}^d \rightarrow \mathbb{R}^d, \boldsymbol{x}_i \mapsto \hat{\boldsymbol{x}}_0 := \frac{1}{a_i}(\boldsymbol{x}_i + b_i^2 s_\theta(\boldsymbol{x}_i, i)).$$

*Then, $Q_i(\boldsymbol{x}_i) \in \mathcal{M}$ and $\boldsymbol{J}_{Q_i}^2 = \boldsymbol{J}_{Q_i} = \boldsymbol{J}_{Q_i}^T : \mathbb{R}^d \rightarrow T_{Q_i(\boldsymbol{x}_i)}\mathcal{M}$. Intuitively, $Q_i$ is locally an orthogonal projection onto $\mathcal{M}$.*

According to the proposition, the score function only concerns the normal direction of the data manifold. In other words, the score function cannot discriminate two data points whose difference is tangent to the manifold. In solving inverse problems, however, we desire to discriminate data points to reconstruct the original signal, and the discrimination is achievable by measurement fidelity. In order to achieve the original signal, the measurement plays a role in correcting the tangent component near the data manifold. Furthermore, with regard to remark 2, diffusion model-based inverse problem solvers should follow the tangent component. The following theorem shows how existing algorithms and the proposed method are different in this regard.

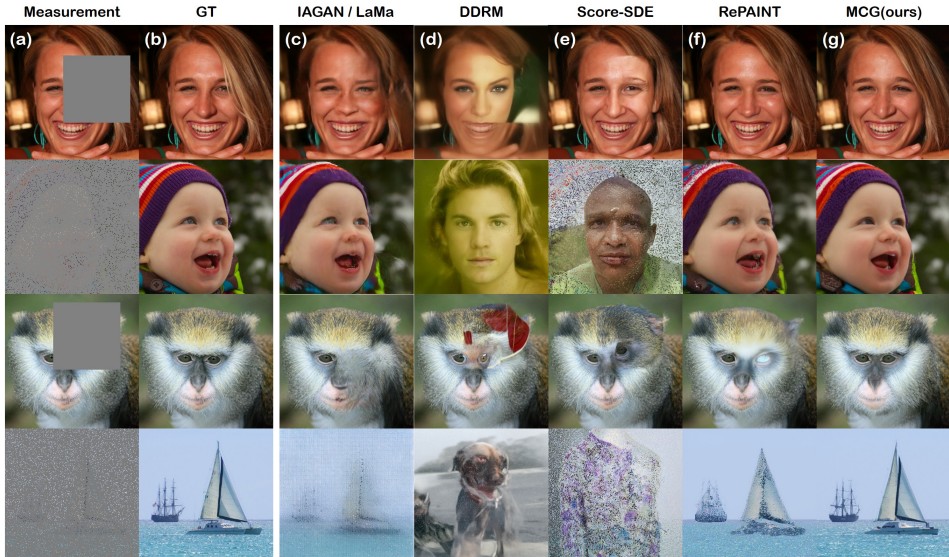

| Measurement | GT | IAGAN / LaMa | DDRM | Score-SDE | RePAINT | MCG(ours) |

Figure 3: Inpainting results on FFHQ (1st, 2nd row) and ImageNet (3rd, 4th row). (a) Measurement, (b) Ground truth, (c) IAGAN [20] for FFHQ, LaMa [43] for ImageNet, (d) DDRM [25], (e) Score-SDE [41], (f) RePAINT [32], (g) MCG (Ours). Out of $256 \times 256$ image, the 1st and the 3rd row is masked with size $128 \times 128$ box. 92% of pixels (all RGB channels) from the images in the 2nd and 4th row are blocked.

**Theorem 1** (Manifold constrained gradient). *A correction by the manifold constrained gradient does not leave the data manifold. Formally,*

$$\frac{\partial}{\partial \boldsymbol{x}_i} \|\boldsymbol{W}(\boldsymbol{y} - \boldsymbol{H}\hat{\boldsymbol{x}}_0)\|_2^2 = -2\boldsymbol{J}_{Q_i}^T \boldsymbol{H}^T \boldsymbol{W}^T \boldsymbol{W}(\boldsymbol{y} - \boldsymbol{H}\hat{\boldsymbol{x}}_0) \in T_{\hat{\boldsymbol{x}}_0}\mathcal{M},$$

*the gradient is the projection of the data fidelity term onto $T_{\hat{\boldsymbol{x}}_0}\mathcal{M}$,*

This theorem suggests that in diffusion models, the naive measurement fidelity step (without considering the data manifold) pushes the inference path out of the manifolds and might lead to inaccurate reconstruction. (To see this pictorially, see section. D, and Fig. 7.) On the other hand, our correction term from the manifold constraint guides the diffusion to lie on the data manifold, leading to better reconstruction. Such geometric views are illustrated in Fig. 2b.

**Remark 3.** *One may concern that the suboptimality of the denoising score matching loss optimization may lead to inaccurate inference of the MCG steps. In practice, however, most of the error in denoising score matching is concentrated on $t \sim 1$[9], and in such region, the Tweedie's inference cannot make meaningful images. That is, the score function cannot detect the data manifold. Nonetheless, in this regime, the magnitudes of the MCGs are small when the denoising score is inaccurate, and hence the matters arising from suboptimality is minimal. As $t \to 0$, the estimation becomes exact, and subsequently leads to accurate implementation of the MCG.*

## 5 Experiments

For all tasks, we aim to verify the superiority of our method against other diffusion model-based approaches, and also against strong supervised learning-based baselines. Further details can be found in Section. F.

**Datasets and Implementation** For inpainting, we use FFHQ $256 \times 256$ [24], and ImageNet $256 \times 256$ [12] to validate our method. We utilize pre-trained models from the open sourced repository based on the implementation of ADM (VP-SDE) [13]. We validate the performance on 1000 held-out validation set images for both FFHQ and ImageNet dataset. For the colorization task, we use FFHQ $256 \times 256$, and LSUN-bedroom $256 \times 256$ [51]. We use pre-trained score functions from

| Method | FFHQ (256 × 256) | | | | | | | | ImageNet (256 × 256) | | | | | |
| | Box | | Random | | Extreme | | Wide masks | | Box | | Random | | Wide masks | |
| | FID ↓ | LPIPS ↓ | FID ↓ | LPIPS ↓ | FID ↓ | LPIPS ↓ | FID ↓ | LPIPS ↓ | FID ↓ | LPIPS ↓ | FID ↓ | LPIPS ↓ | FID ↓ | LPIPS ↓ |
|---|---|---|---|---|---|---|---|---|---|---|---|---|---|---|
| MCG (ours) | **23.7** | 0.089 | **21.4** | **0.186** | **30.6** | **0.366** | 22.1 | 0.099 | **25.4** | 0.157 | **34.8** | **0.308** | 21.9 | 0.148 |
| Score-SDE [41] | 30.3 | 0.135 | 109.3 | 0.674 | 48.6 | 0.488 | 29.8 | 0.132 | 43.5 | 0.199 | 143.5 | 0.758 | 25.9 | 0.150 |
| RePAINT* [32] | 25.7 | 0.093 | 38.1 | 0.240 | 35.9 | 0.398 | 24.2 | 0.108 | 26.1 | 0.156 | 59.3 | 0.387 | 37.0 | 0.205 |
| DDRM [25] | 28.4 | 0.109 | 111.6 | 0.774 | 48.1 | 0.532 | 27.5 | 0.113 | 88.8 | 0.386 | 99.6 | 0.767 | 80.6 | 0.398 |
| LaMa [43] | 27.7 | **0.086** | 188.7 | 0.648 | 61.7 | 0.492 | 23.2 | **0.096** | 26.8 | **0.139** | 134.1 | 0.567 | **20.4** | **0.140** |
| AOT-GAN [52] | 29.2 | 0.108 | 97.2 | 0.514 | 69.5 | 0.452 | 28.3 | 0.106 | 35.3 | 0.163 | 119.6 | 0.583 | 29.8 | 0.161 |
| ICT [49] | 27.3 | 0.103 | 91.3 | 0.445 | 56.7 | 0.425 | 26.9 | 0.104 | 31.9 | 0.148 | 131.4 | 0.584 | 25.4 | 0.148 |
| DSI [35] | 27.9 | 0.096 | 126.4 | 0.601 | 77.5 | 0.463 | 28.3 | 0.102 | 34.5 | 0.155 | 132.9 | 0.549 | 24.3 | 0.154 |
| IAGAN [20] | 26.3 | 0.098 | 41.5 | 0.279 | 56.1 | 0.417 | 23.8 | 0.110 | - | - | - | - | - | - |

Table 1: Quantitative evaluation (FID, LPIPS) of inpainting task on FFHQ and ImageNet. *: Re-implemented with our score function. MCG, Score-SDE, RePAINT, and DDRM all share the same score function and differ only in the inference method. **Bold**: Best, under: second best.

score-SDE [41] based on VE-SDE. We use 300 validation images for testing the performance with respect to the LSUN-bedroom dataset. For experiments with CT, we train our model based on `ncsnpp` as a VE-SDE from score-SDE [41], on the 2016 American Association of Physicists in Medicine (AAPM) grand challenge dataset, and we process the data as in [23]. Specifically, the dataset contains 3839 training images resized to 256×256 resolution. We simulate the CT measurement process with parallel beam geometry with evenly-spaced 180 degrees. Evaluation is performed on 421 held-out validation images from the AAPM challenge.

**Inpainting** Score-SDE [41], REPAINT [32], DDRM [25] were chosen as baseline diffusion models to compare against the proposed method. For a fair comparison, we use the same score function for all methods including MCG, and only differentiate the inference method that is used. Another class of generative models: GAN-based inverse problem solver, IA-GAN [20] is considered as a comparison method for FFHQ specifically. We also include comparisons against supervised learning based baselines: LaMa [43], AOT-GAN [52], ICT [49], and DSI [35]. We use various forms of inpainting masks: box (128 × 128 sized square region is missing[2]), extreme (only the box region is existent), random (90-95% of pixels are missing), and LaMa-wide. Quantitative evaluation is performed with two metrics - Frechet Inception Distance (FID)-1k [17], and Learned Perceptual Image Patch Similarity (LPIPS) [54].

| Data | FFHQ(256×256) | | LSUN(256×256) | |
| Method | SSIM ↑ | LPIPS ↓ | SSIM ↓ | LPIPS ↓ |
|---|---|---|---|---|
| MCG (ours) | 0.951 | **0.146** | **0.959** | **0.160** |
| Score-SDE [41] | 0.936 | 0.180 | 0.945 | 0.199 |
| DDRM [25] | 0.948 | 0.154 | 0.957 | 0.182 |
| cINN [2] | **0.952** | 0.166 | 0.952 | 0.180 |
| pix2pix [21] | 0.935 | 0.184 | 0.947 | 0.174 |

Table 2: Quantitative evaluation (SSIM, LPIPS) of colorization task. **Bold**: best, under: second best.

Our method outperforms the diffusion model baselines [41, 32, 25] by a large margin. Moreover, our method is also competitive with, or even better than the best-in-class fully supervised methods, as can be seen in Table 1. In Fig. 3, we depict representative results that show the superiority of the method, where we see in both the box-type and random dropping that MCG performs very well on all experiments.

| | AAPM (256 × 256) | | | |
| Views | 18 | | 30 | |
| Method | PSNR ↑ | SSIM ↑ | PSNR ↑ | SSIM ↑ |
|---|---|---|---|---|
| MCG (ours) | **33.57** | **0.956** | **36.09** | **0.971** |
| Score-CT [40] | 29.85 | 0.897 | 31.97 | 0.913 |
| SIN-4c-PRN [50] | 26.96 | 0.850 | 30.23 | 0.917 |
| cGAN [15] | 24.38 | 0.823 | 27.45 | 0.927 |
| FISTA-TV [3] | 21.57 | 0.791 | 23.92 | 0.861 |

Table 3: Quantitative evaluation (PSNR, SSIM) of CT reconstruction task. **Bold**: best.

**Colorization** We choose score-SDE [41], and DDRM [25] as diffusion-model based comparison methods, and also compare against cINN [2], and pix2pix [21]. Two metrics were used for evaluation: structural similarity index (SSIM), and LPIPS. Consistent with the findings from inpainting, we achieve much improved performance than score-SDE, and also is favorable against state-of-the-art (SOTA) superivsed learning based methods. In Table 2, we see that the proposed

---

[2]The location of the box is sampled uniformly within 16 pixel margin of each side.

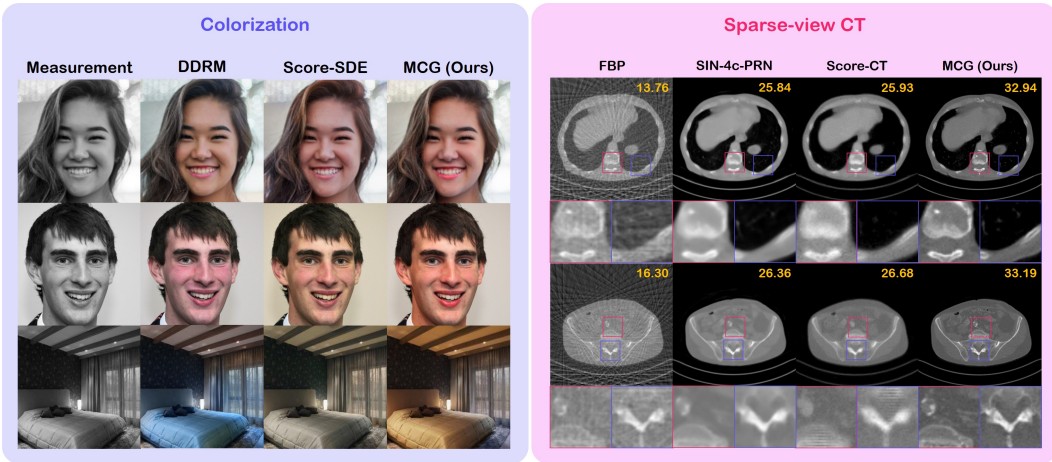

Figure 4: Colorization results on FFHQ / LSUN-bedroom, Sparse view CT reconstruction results on AAPM.

method outperforms all other methods in terms of both PSNR/LPIPS in LSUN-bedroom, and also achieves strong performance in the colorization of FFHQ dataset.

**CT reconstruction**    To the best of our knowledge, [40] is the only method that tackles CT reconstruction directly with diffusion models. We compare our method against [40], which we refer to as score-CT henceforth. We also compare with the best-in-class supervised learning methods, cGAN [15] and SIN-4c-PRN [50]. As a compressed sensing baseline, FISTA-TV [3] was included, along with the analytical reconstruction method, FBP. We use two standard metrics - peak-signal-to-noise-ratio (PSNR), and SSIM for quantitative evaluation. From Table 3, we see that the newly proposed MCG method outperforms the previous score-CT [40] by a large margin. We can observe the superiority of MCG over other methods more clearly in Fig. 4, where MCG reconstructs the measurement with high fidelity and detail. All other methods including the fully supervised baselines fall behind the proposed method.

**Ablation studies**    We perform three ablation studies: 1) As both the MCG term and the projection term contain information about the measurement $y$, we observe the contribution of each term to the fixed solution. To further clarify the efficacy of the gradient step combined with Tweedie's denoising, we also consider the case where the gradient of the log likelihood is computed not in the noiseless regime, but in the noise level matching the current iteration. Specifically, we define $x'_{i-1} := f(x_i, s_\theta) + g(x_i)z$, $z \sim \mathcal{N}(0, I)$ , $y_{i-1} \sim p(y_{i-1}|y_0)$, and implement the gradient step as $\nabla_{x_i} \|y_{i-1} - Hx'_{i-1}\|_2^2$. 2) As the performance of diffusion models depend heavily on the number of NFEs, we observe the trade-off of each diffusion model when varying the NFE from 20 to 1000. Moreover, for completeness, we measure the runtime of each algorithms including the non-diffusion based methods in wall-clock time computed with a commodity GPU in Table. 4. 3) Setting $\alpha = 0.0$ reduces our method to [9]. We show the difference in the performance by varying the values of $\alpha$.

| Method | Wall-clock time [s] |
|---|---|
| Score-SDE [41] | 38.68 |
| RePAINT [32] | 247.6 |
| DDRM [25] | 2.117 |
| LaMa [43] | 0.629 |
| AOT-GAN [52] | 0.082 |
| ICT [49] | 144.6 |
| DSI [35] | 36.64 |
| IAGAN [20] | 518.47 |
| Ours | 81.59 |

Table 4: Runtime for each algorithm in Wall-clock time: Computed with a single GTX 1080Ti GPU.

First, we see in Table. 5 that using only the MCG step leads to improved performance in terms of LPIPS, but introduces error in the measurement consistency (measured with MSE). Combining both the projection and MCG leads to perfect data consistency along with further improved

| Method | LPIPS($\downarrow$) | MSE(MC) |
|---|---|---|
| Proj. | 0.138 | 0 |
| $\nabla_{x_i} \|y_{i-1} - Hx'_{i-1}\|_2^2$ | 0.271 | 12.99 |
| $\nabla_{x_i} \|y_{i-1} - Hx'_{i-1}\|_2^2$ + Proj. | 0.128 | 0 |
| $\nabla_{x_i} \|y - H\hat{x}_0\|_2^2$ | 0.124 | 10.7 |
| $\nabla_{x_i} \|y - H\hat{x}_0\|_2^2$ + Proj. (**Ours**) | **0.089** | **0** |

Table 5: LPIPS & Measurement consistency (MC) vs. method

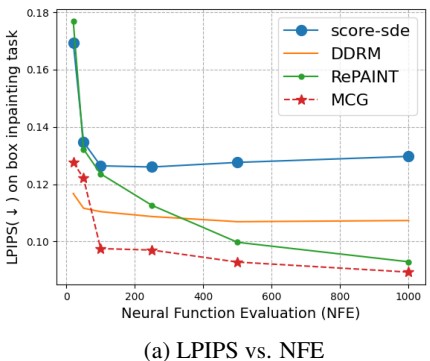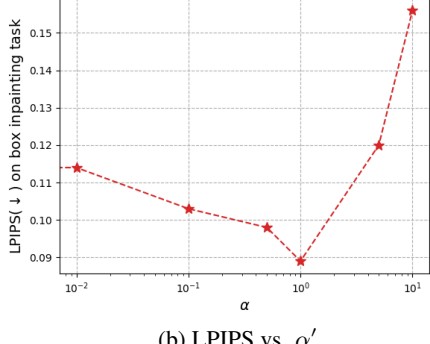

(a) LPIPS vs. NFE

(b) LPIPS vs. $\alpha'$

Figure 5: Ablation studies performed with box inpainting task on FFHQ 256×256 data.

reconstruction. When considering gradient steps without Tweedie's denoising (i.e. keeping the noise level at the $i^{\text{th}}$ step), the performance heavily degrades, especially when implemented without the projection steps. Here, we see that the proposed denoising step to utilize $\hat{x}_0$ is indeed the key to the superior performance.

Second, looking at Fig. 5a, we immediately see that the graph of MCG stays in the lowest (best) LPIPS regime across all NFEs by a large margin, except for when the NFE drops below 100. Here, DDRM [25] takes over the 1st place - allegedly due to the DDIM sampling strategy they take. The performance of RePAINT deteriorates rapidly as we decrease NFE. Furthermore, we observe that the LPIPS of score-SDE [41] actually *increases* (i.e. worsen), as we increase the number of NFEs from a few hundred to one thousand. This suggests that the inference process that score-SDE takes (i.e. projection only) is inherently flawed, and cannot be corrected by taking small enough steps. In Table. 4, we list the runtime of all the methods that were used for comparison in the task of inpainting. Note that the proposed method takes longer for compute than score-SDE albeit having the same NFE. The gap is due to the backpropagation steps that are required for the MCG step, where the gap can be potentially ameliorated by switching to JAX [6] implementation from the current PyTorch implementation.

Lastly, we observe the difference in the performance as we vary the values of $\alpha$. Implementation-wise, we find that we yield superior results when normalizing the squared norm with the norm of itself (e.g. $\alpha = \alpha'/\|\boldsymbol{W}(\boldsymbol{y} - \boldsymbol{H}\hat{\boldsymbol{x}}_0)\|$, where $\alpha'$ is some constant). In order to avoid cluttered notation, we instead experiment with changing the values of $\alpha'$ in Fig. 5b. Inspecting Fig. 5b, we see that $\alpha$ values within the range $[0.1, 1.0]$ produce satisfactory results. $\alpha$ values that are too low do not fully enjoy the advantages of MCG and collapses to the projection-only method, while using too high values of $\alpha$ results in exploding gradients, and the reconstruction saturates.

**Properties of our method**   Our proposed method is fully unsupervised and is not trained on solving a specific inverse problem. For example, our box masks and random masks have very different forms of erasing the pixel values. Nevertheless, our method generalizes perfectly well to such different measurement conditions, while other methods have a large performance gap between the different mask shapes. We further note two appealing properties of our method as an inverse problem solver: 1) the ability to generate multiple solutions given a condition, and 2) the ability to maintain perfect measurement consistency. The former ability often lacks in supervised learning-based methods [43, 50], and the latter is often not satisfied for some unsupervised GAN-based solutions [10, 4].

## 6   Conclusion

In this work, we proposed a general framework that can greatly enhance the performance of the diffusion model-based solvers for solving inverse problems. We showed several promising applications - inpainting, colorization, sparse view CT reconstruction, and showed that our method can outperform the current state-of-the-art methods. We analyzed our method theoretically and show that

MCG prevents the data generation process from falling off the manifold, thereby reducing the errors that might accumulate at every step. Further, we showed that MCG controls the direction tangent to the data manifold, whereas the score function controls the direction that is normal, such that the two components complement each other.

**Limitations and Broader Impact**    The proposed method is inherently stochastic since the diffusion model is the main workhorse of the algorithm. When the dimension $m$ is pushed to low values, at times, our method fails to produce high quality reconstructions, albeit being better than the other methods overall. For extreme cases of inpainting (e.g. Half masks) with the ImageNet model, we often observe artifacts in our reconstruction (e.g. generating perfectly symmetric images), which we discuss in further detail in Sec. E. We note that our method is slow to sample from, inheriting the existing limitations of diffusion models. This would likely benefit from leveraging recent solvers aimed at accelerating the inference speed of diffusion models. In line with the arguments of other generative model-based inverse problem solvers, our method is a solver that relies heavily on the underlying diffusion model, and can thus potentially create malicious content such as deepfakes. Further, the reconstructions could intensify the social bias that is already existent in the training dataset.

## Acknowledgments and Disclosure of Funding

This research was supported by Field-oriented Technology Development Project for Customs Administration through National Research Foundation of Korea(NRF) funded by the Ministry of Science & ICT and Korea Customs Service (NRF-2021M3I1A1097938, NRF-2021M3I1A1097910), by the Korea Health Technology R&D Project through the Korea Health Industry Development Institute (KHIDI), which is funded by the Ministry of Health & Welfare, Republic of Korea (grant number: HU21C0222), and by the KAIST Key Research Institute (Interdisciplinary Research Group) Project.

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
