}$, $\boldsymbol{z} \sim \mathcal{N}(0, \boldsymbol{I})$, $\boldsymbol{y}_{i-1} \sim p(\boldsymbol{y}_{i-1}|\boldsymbol{y}_0)$, and implement the gradient step as $\nabla_{\boldsymbol{x}_i} \|\boldsymbol{y}_{i-1} - \boldsymbol{H}\boldsymbol{x}'_{i-1}\|_2^2$. 2) As the performance of diffusion models depend heavily on the number of NFEs, we observe the trade-off of each diffusion model when varying the NFE from 20 to 1000. Moreover, for completeness, we measure the runtime of each algorithms including the non-diffusion based methods in wall-clock time computed with a commodity GPU in Table. 4. 3) Setting $\alpha = 0.0$ reduces our method to [9]. We show the difference in the performance by varying the values of $\alpha$.

| Method | Wall-clock time [s] |
|---|---|
| Score-SDE [41] | 38.68 |
| RePAINT [32] | 247.6 |
| DDRM [25] | 2.117 |
| LaMa [43] | 0.629 |
| AOT-GAN [52] | 0.082 |
| ICT [49] | 144.6 |
| DSI [35] | 36.64 |
| IAGAN [20] | 518.47 |
| Ours | 81.59 |

Table 4: Runtime for each algorithm in Wall-clock time: Computed with a single GTX 1080Ti GPU.

First, we see in Table. 5 that using only the MCG step leads to improved performance in terms of LPIPS, but introduces error in the measurement consistency (measured with MSE). Combining both the projection and MCG leads to perfect data consistency along with further improved

| Method | LPIPS($\downarrow$) | MSE(MC) |
|---|---|---|
| Proj. | 0.138 | 0 |
| $\nabla_{\boldsymbol{x}_i}\|\boldsymbol{y}_{i-1} - \boldsymbol{H}\boldsymbol{x}'_{i-1}\|_2^2$ | 0.271 | 12.99 |
| $\nabla_{\boldsymbol{x}_i}\|\boldsymbol{y}_{i-1} - \boldsymbol{H}\boldsymbol{x}'_{i-1}\|_2^2$ + Proj. | 0.128 | 0 |
| $\nabla_{\boldsymbol{x}_i}\|\boldsymbol{y} - \boldsymbol{H}\hat{\boldsymbol{x}}_0\|_2^2$ | 0.124 | 10.7 |

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

*Proof.* Suppose that the data manifold is an $l$-dimensional linear subspace. By rotation and translation, we safely assume that $\mathcal{M} = \{\boldsymbol{x} \in \mathbb{R}^n : x_{l+1} = x_{l+2} = \cdots = x_n = 0\}$. Then, we can simply write $d(\boldsymbol{x}, \mathcal{M}) = \sqrt{x_{l+1}^2 + \cdots + x_n^2}$, and $\mathcal{M}_i = \{\boldsymbol{x} \in \mathbb{R}^n : x_{l+1}^2 + \cdots + x_n^2 = r_i^2\}$. For a given point $\boldsymbol{x}' = (x_1', x_2', \dots) \in \mathcal{M}$, we consider $p(\boldsymbol{x}|\boldsymbol{x}') \sim \mathcal{N}(a_i\boldsymbol{x}', b_i^2 I)$ and obtain a concentration inequality independent to the choice of $\boldsymbol{x}'$. We need the standard Laurent-Massart bound for a chi-square variable [31]. When $X$ is a chi-square distribution with $k$ degrees of freedom,

$$P[X - k \geq 2\sqrt{kt} + 2t] \leq e^{-t},$$
$$P[X - k \leq -2\sqrt{kt}] \leq e^{-t}.$$

As $\frac{x_{l+1}^2}{b_i^2} + \cdots + \frac{x_n^2}{b_i^2}$ is a chi-square distribution with $n - l$ degrees of freedom, by substituting $t = (n-l)\varepsilon'$ in the above bound,

$$P\left[-2(n-l)\sqrt{\varepsilon'} \leq \frac{x_{l+1}^2}{b_i^2} + \cdots + \frac{x_n^2}{b_i^2} - (n-l) \leq 2(n-l)(\sqrt{\varepsilon'} + \varepsilon')\right]$$
$$= P\left[\sqrt{x_{l+1}^2 + \cdots + x_n^2} \in (r_i\sqrt{1 - 2\sqrt{\varepsilon'}}, r_i\sqrt{1 + 2\sqrt{\varepsilon'} + 2\varepsilon'})\right] \geq 1 - 2e^{-(n-l)\varepsilon'}.$$

Note that the above inequality does not depend on $x_1, \dots x_l$, thus the choice of $\boldsymbol{x}' \in \mathcal{M}$. As a result, by setting $\varepsilon = \min\{1 - \sqrt{1 - 2\sqrt{\varepsilon'}}, \sqrt{1 + 2\sqrt{\varepsilon'} + 2\varepsilon'} - 1\}$ and $\delta = 2e^{-(n-l)\varepsilon'}$,

$$p(\boldsymbol{x} \in B_{\varepsilon r_i}(M_i)|\boldsymbol{x}') > 1 - \delta,$$

thus

$$p_i(\boldsymbol{x} \in B_{\varepsilon r_i}(M_i)) = \int p(\boldsymbol{x} \in B_{\varepsilon r_i}(M_i)|\boldsymbol{x}')p(\boldsymbol{x}')d\boldsymbol{x}' > 1 - \delta.$$

$\square$

**Proposition 2** (score function). *Suppose $s_\theta$ is the minimizer of the denoising score matching loss in (3). Let $Q_i$ be the function that maps $\boldsymbol{x}_i$ to $\hat{\boldsymbol{x}}_0$ for each $i$,*

$$Q_i : \mathbb{R}^d \to \mathbb{R}^d, \boldsymbol{x}_i \mapsto \hat{\boldsymbol{x}}_0 := \frac{1}{a_i}(\boldsymbol{x}_i + b_i^2 s_\theta(\boldsymbol{x}_i, i)).$$

*Then, $Q_i(\boldsymbol{x}_i) \in \mathcal{M}$ and $\boldsymbol{J}_{Q_i}^2 = \boldsymbol{J}_{Q_i} = \boldsymbol{J}_{Q_i}^T : \mathbb{R}^d \to T_{Q_i(\boldsymbol{x}_i)}\mathcal{M}$. Intuitively, $Q_i$ is locally an orthogonal projection onto $\mathcal{M}$.*

*Proof.* To minimize (3), or equivalently,

$$\int ||s_\theta(\boldsymbol{x}_t, t) - \nabla_{\boldsymbol{x}_t} \log p(\boldsymbol{x}_t|\boldsymbol{x}_0)||_2^2 p(\boldsymbol{x}_t|\boldsymbol{x})p(\boldsymbol{x})d\boldsymbol{x}d\boldsymbol{x}_t dt,$$

By differentiating the objective with respect to $s_\theta(\boldsymbol{x}_t, t)$, we have

$$\int \left( s_\theta(\boldsymbol{x}_t, t) - \frac{a_t \boldsymbol{x} - \boldsymbol{x}_t}{b_t^2} \right) p(\boldsymbol{x}_t|\boldsymbol{x})p(\boldsymbol{x})d\boldsymbol{x} = 0$$

$$\int s_\theta(\boldsymbol{x}_t, t)p(\boldsymbol{x}_t)p(\boldsymbol{x}|\boldsymbol{x}_t)d\boldsymbol{x} = \int \frac{a_t \boldsymbol{x} - \boldsymbol{x}_t}{b_t^2} p(\boldsymbol{x}_t)p(\boldsymbol{x}|\boldsymbol{x}_t)d\boldsymbol{x}$$

$$s_\theta(\boldsymbol{x}_t, t) \int p(\boldsymbol{x}|\boldsymbol{x}_t)d\boldsymbol{x} = \int \frac{a_t \boldsymbol{x}}{b_t^2} p(\boldsymbol{x}|\boldsymbol{x}_t)d\boldsymbol{x} - \frac{\boldsymbol{x}_t}{b_t^2} \int p(\boldsymbol{x}|\boldsymbol{x}_t)d\boldsymbol{x}$$

$$\therefore s_\theta(\boldsymbol{x}_t, t) = \frac{1}{b_t^2}(-\boldsymbol{x}_t + a_t \int \boldsymbol{x}p(\boldsymbol{x}|\boldsymbol{x}_t)d\boldsymbol{x})\forall \boldsymbol{x}_t, t,$$

where we used $p(\boldsymbol{x}_t|\boldsymbol{x})p(\boldsymbol{x}) = p(\boldsymbol{x}, \boldsymbol{x}_t) = p(\boldsymbol{x}_t)p(\boldsymbol{x}|\boldsymbol{x}_t)$, $p(\boldsymbol{x}_t) > 0$, and $\int p(\boldsymbol{x}|\boldsymbol{x}_t)d\boldsymbol{x} = 1$ in each line. Here, $Q_i(\boldsymbol{x}_i) = \int \boldsymbol{x}p(\boldsymbol{x}|\boldsymbol{x}_i)d\boldsymbol{x}$ is the weighted average vector of points on the data manifold as $p(\boldsymbol{x}|\boldsymbol{x}_i)$ is supported on the data manifold. Combining it with the assumption that the manifold is linear, $Q_i(\boldsymbol{x}_i) \in \mathcal{M}$.

Considering the symmetry of $p(\boldsymbol{x}|\boldsymbol{x}_i)$ about $\boldsymbol{x}_i$, $p(\boldsymbol{x}|\boldsymbol{x}_i)$ is a radial function on $\mathcal{M}$, centering around the nearest point to $\boldsymbol{x}_i$ on $\mathcal{M}$. Hence, $Q_i(\boldsymbol{x}_i)$ shall be the nearest point to $\boldsymbol{x}_i$ of all points on $\mathcal{M}$. Therefore, $J_{Q_i}$ is the orthogonal projection onto $T_{Q_i(\boldsymbol{x}_i)}\mathcal{M}$. Stating more rigorously, let $\boldsymbol{u} = \boldsymbol{u}_t + \boldsymbol{u}_n \in \mathbb{R}^n$ for $\boldsymbol{u}_t \in T_{Q_i(\boldsymbol{x}_i)}\mathcal{M}, \boldsymbol{u}_n \perp T_{Q_i(\boldsymbol{x}_i)}\mathcal{M}$. Then, for a scalar $s$, $Q_i(\boldsymbol{x}_i + s\boldsymbol{u}) = Q_i(\boldsymbol{x}_i) + s\boldsymbol{u}_t$, as only tangent component to the manifold change the nearest point. By differentiating with respect to $s$, we obtain $\boldsymbol{J}_{Q_i}\boldsymbol{u} = \boldsymbol{u}_t$, thus $\boldsymbol{J}_{Q_i}^2 = \boldsymbol{J}_{Q_i}$. For another vector $\boldsymbol{v} = \boldsymbol{v}_t + \boldsymbol{v}_n$ with $\boldsymbol{v}_t \in T_{Q_i(\boldsymbol{x}_i)}\mathcal{M}, \boldsymbol{v}_n \perp T_{Q_i(\boldsymbol{x}_i)}\mathcal{M}$,

$$\boldsymbol{v}^T \boldsymbol{J}_{Q_i}\boldsymbol{u} = (\boldsymbol{v}_t + \boldsymbol{v}_n)^T \boldsymbol{u}_t$$
$$= \boldsymbol{v}_t^T \boldsymbol{u}_t$$
$$= (\boldsymbol{u}_t + \boldsymbol{u}_n)^T \boldsymbol{v}_t$$
$$= \boldsymbol{u}^T \boldsymbol{J}_{Q_i}\boldsymbol{v},$$

where we applied $\boldsymbol{v}_n^T \boldsymbol{u}_t = 0 = \boldsymbol{u}_n^T \boldsymbol{v}_t$. Therefore, $\boldsymbol{J}_{Q_i}$ is symmetric, i.e. $\boldsymbol{J}_{Q_i}^T = \boldsymbol{J}_{Q_i}$, which concludes this proof. $\qquad\square$

**Theorem 1** (Manifold constrained gradient). *A correction by the manifold constrained gradient does not leave the data manifold. Formally,*

$$\frac{\partial}{\partial \boldsymbol{x}_i} \|\boldsymbol{W}(\boldsymbol{y} - \boldsymbol{H}\hat{\boldsymbol{x}}_0)\|_2^2 = -2\boldsymbol{J}_{Q_i}^T \boldsymbol{H}^T \boldsymbol{W}^T \boldsymbol{W}(\boldsymbol{y} - \boldsymbol{H}\hat{\boldsymbol{x}}_0) \in T_{\hat{\boldsymbol{x}}_0}\mathcal{M},$$

*the gradient is the projection of the data fidelity term onto $T_{\hat{\boldsymbol{x}}_0}\mathcal{M}$,*

*Proof.*

$$\frac{\partial}{\partial \boldsymbol{x}_i} \|\boldsymbol{W}(\boldsymbol{y} - \boldsymbol{H}\hat{\boldsymbol{x}}_0)\|_2^2 = -2\boldsymbol{J}_{\boldsymbol{W}\boldsymbol{H}Q_i}^T \boldsymbol{W}(\boldsymbol{y} - \boldsymbol{H}\hat{\boldsymbol{x}}_0)$$
$$= -2\boldsymbol{J}_{Q_i}^T \boldsymbol{H}^T \boldsymbol{W}^T \boldsymbol{W}(\boldsymbol{y} - \boldsymbol{H}\hat{\boldsymbol{x}}_0)$$
$$= \boldsymbol{J}_{Q_i}d \in T_{Q_i(\boldsymbol{x}_i)}\mathcal{M}$$

where $d = -2\boldsymbol{H}^T \boldsymbol{W}^T \boldsymbol{W}(\boldsymbol{y} - \boldsymbol{H}\hat{\boldsymbol{x}}_0)$. The first and second equality is given by the chain rule and the last line is by Proposition 2. $\qquad\square$

In Fig 6, we illustrate how the proposed algorithm benefits from mixing the MCG step with the conventional POCS step. Pushing the points to the tangent directions, we expect less deviation from the manifold which is attributed to POCS.

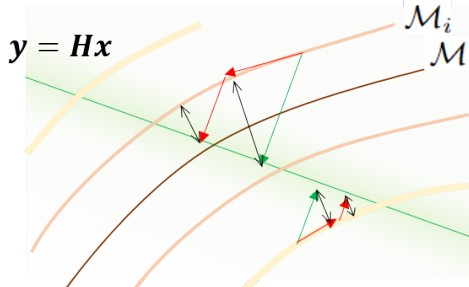

Figure 6: The advantage of mixing the MCG and the POCS steps over the conventional POCS step. Each curve represents a manifold of (noisy) data. Arrows suggest the POCS steps (green arrows) and steps mixing the MCG and the POCS (red arrows). Due to the path along the manifolds, proposed mixing step alleviates reverse diffusion step leaving the manifolds (black arrows).

## B  Discrete forms of SDE

Here, we review the different types of SDEs and sampling algorithms that we use throughout the paper for completeness. We assume that the time horizon $[0, 1]$ is linearly split up into $N$ discretization segments, such that all intervals have the length $1/N$, if not specified otherwise.

### B.1  Forward diffusion

Due to the linearity of the drift and diffusion functions, we can analytically sample from $p(\boldsymbol{x}_i|\boldsymbol{x}_0)$ via reparameterization trick:

$$\boldsymbol{x}_i = a_i\boldsymbol{x}_0 + b_i\boldsymbol{z}, \ \boldsymbol{z} \sim \mathcal{N}(0, \boldsymbol{I}). \tag{16}$$

In VP-SDE [18], one defines a linearly increasing noise schedule $\beta_1, \beta_2, \ldots, \beta_N \in (0, 1)$. Further, we define $\alpha_i = 1 - \beta_i$, and $\bar{\alpha}_i = \prod_{j=1}^{i} \alpha_j$. Then, the forward diffusion process can be implemented as

$$\boldsymbol{x}_i = \sqrt{\bar{\alpha}_i}\boldsymbol{x}_0 + \sqrt{1 - \bar{\alpha}_i}\boldsymbol{z}, \ \boldsymbol{z} \sim \mathcal{N}(0, \boldsymbol{I}). \tag{17}$$

In VE-SDE [41], one defines a geometrically increasing noise schedule $\sigma_i = \sigma_0 \left(\frac{\sigma_N}{\sigma_0}\right)^{\frac{i-1}{N-1}}$. Since the drift function is zero, the forward diffusion simply becomes Brownian motion. Concretely,

$$\boldsymbol{x}_i = \boldsymbol{x}_0 + \sigma_i\boldsymbol{z}, \ \boldsymbol{z} \sim \mathcal{N}(0, \boldsymbol{I}). \tag{18}$$

### B.2  Reverse diffusion

First, for the case of VP-SDE, the reverse diffusion step is implemented by

$$\boldsymbol{x}_{i-1} = \frac{1}{\sqrt{\alpha_i}} \left(\boldsymbol{x}_i - \frac{1 - \alpha_i}{\sqrt{1 - \bar{\alpha}_i}}\boldsymbol{z}_\theta(\boldsymbol{x}_i, i)\right) + \sqrt{\tilde{\sigma}_i}\boldsymbol{z}, \ \ \boldsymbol{z} \sim \mathcal{N}(0, \boldsymbol{I}), \tag{19}$$

where $\boldsymbol{z}_\theta(\boldsymbol{x}_i, i)$ is trained with the epsilon-matching scheme as in [18], and $\tilde{\sigma}_i$ is set to a learnable parameter as in [13]. Note that eq. (19) was written in terms of $\boldsymbol{z}_\theta(\boldsymbol{x}_i, i)$ and not in terms of the score function, $\boldsymbol{s}_\theta(\boldsymbol{x}_i, i)$. One can re-write the expression using the relation $\boldsymbol{z}_\theta(\boldsymbol{x}_i, i) = -\sqrt{1 - \bar{\alpha}_i}\boldsymbol{s}_\theta(\boldsymbol{x}_i, i)$, as

$$\boldsymbol{x}_{i-1} = \frac{1}{\sqrt{\alpha_i}} \left(\boldsymbol{x}_i + (1 - \alpha_i)\boldsymbol{s}_\theta(\boldsymbol{x}_i, i)\right) + \sqrt{\sigma_i}\boldsymbol{z}, \ \boldsymbol{z} \sim \mathcal{N}(0, \boldsymbol{I}). \tag{20}$$

Next, for the VE-SDE, the reverse diffusion step using the Euler-Maruyama solver [38] is given as

$$\boldsymbol{x}_{i-1} = \boldsymbol{x}_i + (\sigma_i^2 - \sigma_{i-1}^2)\boldsymbol{s}_\theta(\boldsymbol{x}_i, i) + \sqrt{\sigma_i^2 - \sigma_{i-1}^2}\boldsymbol{z}, \ \boldsymbol{z} \sim \mathcal{N}(0, \boldsymbol{I}). \tag{21}$$

Summary is presented in Table 6.

| Type | $a_i$ | $b_i$ | $\boldsymbol{f}(\boldsymbol{x}_i, s_\theta)$ | $g(i)$ |
|---|---|---|---|---|
| VP-SDE | $\sqrt{\bar{\alpha}_i}$ | $\sqrt{1-\bar{\alpha}_i}$ | $\frac{1}{\sqrt{\bar{\alpha}_i}}\left(\boldsymbol{x}_i + (1-\alpha_i)s_\theta(\boldsymbol{x}_i, i)\right)$ | $\sqrt{\tilde{\sigma}_i}$ |
| VE-SDE | $1$ | $\sigma_i$ | $\boldsymbol{x}_i + (\sigma_i^2 - \sigma_{i-1}^2)s_\theta(\boldsymbol{x}_i, i)$ | $\sqrt{\sigma_i^2 - \sigma_{i-1}^2}$ |

Table 6: Choice of $a_i, b_i, \boldsymbol{f}, g$ for each SDE realization.

---

**Algorithm 1** Inpainting (VP, AS)

---

**Require:** $\boldsymbol{y}, \boldsymbol{P}, \{\alpha_i\}_{i=1}^N, \{\tilde{\sigma}_i\}_{i=1}^N, s_\theta, \alpha$
1: $\boldsymbol{x}_N \sim \mathcal{N}(\boldsymbol{0}, \boldsymbol{I})$        ▷ Initial sampling
2: **for** $i = N$ to 1 **do**        ▷ Reverse diffusion
3:      $s \leftarrow s_\theta(\boldsymbol{x}_i, i)$        ▷ Cache score function output
4:      $\boldsymbol{x}'_{i-1} \leftarrow \frac{1}{\sqrt{\alpha_i}}(\boldsymbol{x}_i + (1-\alpha_i)s)$
5:      $\boldsymbol{z} \sim \mathcal{N}(\boldsymbol{0}, \boldsymbol{I})$
6:      $\boldsymbol{x}_{i-1} \leftarrow \boldsymbol{x}'_{i-1} + \tilde{\sigma}_i \boldsymbol{z}$        ▷ Unconditional update
7:      $\boldsymbol{z} \sim \mathcal{N}(\boldsymbol{0}, \boldsymbol{I})$
8:      $\hat{\boldsymbol{x}}_0 \leftarrow \frac{1}{\sqrt{\bar{\alpha}_i}}(\boldsymbol{x}_i + (1-\bar{\alpha}_i)s)$        ▷ $\hat{\boldsymbol{x}}_0$ prediction
9:      $\boldsymbol{y}_i \leftarrow \sqrt{\bar{\alpha}_i}\boldsymbol{y} + \sqrt{1-\bar{\alpha}_i}\boldsymbol{z}$
10:      $\boldsymbol{x}''_{i-1} \leftarrow \boldsymbol{x}'_{i-1} - \alpha\frac{\partial}{\partial \boldsymbol{x}_i}\|\boldsymbol{y} - \boldsymbol{P}\hat{\boldsymbol{x}}_0\|_2^2$        ▷ MCG
11:      $\boldsymbol{z} \sim \mathcal{N}(\boldsymbol{0}, \boldsymbol{I})$
12:      $\boldsymbol{y}_i \leftarrow \sqrt{\bar{\alpha}_i}\boldsymbol{y} + \sqrt{1-\bar{\alpha}_i}\boldsymbol{z}$
13:      $\boldsymbol{x}_{i-1} \leftarrow (\boldsymbol{I} - \boldsymbol{P}^T\boldsymbol{P})\boldsymbol{x}''_{i-1} + \boldsymbol{P}^T\boldsymbol{y}_i$        ▷ Data consistency
14: **end for**
15: **return** $\boldsymbol{x}_0$

---

## C Algorithms

**Inpainting** The forward model for inpainting is given as

$$\boldsymbol{y} = \boldsymbol{P}\boldsymbol{x} + \boldsymbol{\epsilon}, \quad \boldsymbol{P} \in \mathbb{R}^{m\times n}, \tag{22}$$

where $\boldsymbol{P} \in \{0,1\}^{m\times n}$ is the matrix consisting of the columns with standard coordinate vectors indicating the indices of measurement. For the steps in (14), (15), we choose the following

$$\boldsymbol{W} = \boldsymbol{I}, \quad \boldsymbol{A} = \boldsymbol{I} - \boldsymbol{P}^T\boldsymbol{P}, \quad \boldsymbol{b}_i = \boldsymbol{P}^T\boldsymbol{y}_i, \quad \boldsymbol{y}_i \sim q(\boldsymbol{y}_i|\boldsymbol{y}) := \mathcal{N}(\boldsymbol{y}_i|a_i\boldsymbol{y}, b_i^2\boldsymbol{I}). \tag{23}$$

Specifically, $\boldsymbol{A}$ takes the orthogonal complement of $\boldsymbol{x}'_{i-1}$, meaning that the measurement subspace is corrected by $\boldsymbol{y}_i$, while the orthogonal components are updated from $\boldsymbol{x}'_{i-1}$. Note that we use $\boldsymbol{y}_i$ sampled from $\boldsymbol{y}$ to match the noise level of the current estimate.

We provide the algorithm used for inpainting in Algorithm. 1. The sampler is based on basic ancestral sampling (AS) of [18], and the default configuration requires $N = 1000$, $\alpha = 1.0/\|\boldsymbol{y} - \boldsymbol{P}\hat{\boldsymbol{x}}_0\|$ for sampling.

**Colorization** The forward model for colorization is specified as

$$\boldsymbol{y} = \boldsymbol{C}\boldsymbol{x} + \boldsymbol{\epsilon} := \boldsymbol{P}\boldsymbol{M}\boldsymbol{x} + \boldsymbol{\epsilon}, \quad \boldsymbol{P} \in \mathbb{R}^{m\times n}, \quad \boldsymbol{M} \in \mathbb{R}^{n\times n}, \tag{24}$$

where $\boldsymbol{P}$ is the matrix that was used in inpainting, and $\boldsymbol{M}$ is an orthogonal matrix that couples the RGB colormaps[3]. $\boldsymbol{M}^T$ is a matrix that de-couples the channels back to the original space. In other words, one can view colorization as performing imputation in some spectral space. Subsequently, for our colorization method we choose

$$\boldsymbol{W} = \boldsymbol{C}^T, \quad \boldsymbol{A} = \boldsymbol{I} - \boldsymbol{C}^T\boldsymbol{C}, \quad \boldsymbol{b}_i = \boldsymbol{C}^T\boldsymbol{y}_i, \quad \boldsymbol{y}_i \sim q(\boldsymbol{y}_i|\boldsymbol{y}). \tag{25}$$

Again, our forward measurement matrix is orthogonal, and we choose $\boldsymbol{A}$ such that we only affect the orthogonal complement of the measurement subspace.

---

[3]The matrix $\boldsymbol{M}$ is adopted from the colorization matrix of [41].

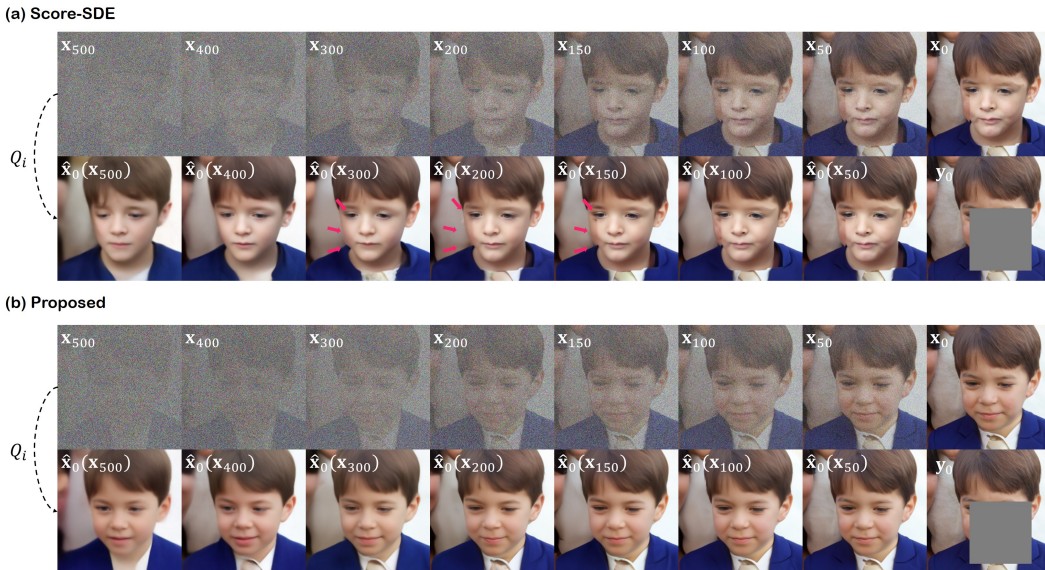

Figure 7: Comparison of the evolution (i.e. generative path) between score-SDE [41], and our method. First rows in (a),(b): Evolution of $x_i$, second rows in (a),(b): Evolution of $\hat{x}_0$.

The sampler for colorization is based on the predictor-corrector (PC) sampler of [41] (VE-SDE), and we choose to apply MCG after every iteration of both predictor, and corrector steps. $N = 2000, \alpha = 0.1/\|C^T(y - C\hat{x}_0)\|$ are chosen as hyper-parameters.

**CT Reconstruction**   For the case of CT reconstruction, the forward model reads

$$y = Rx + \epsilon, \quad R \in \mathbb{R}^{m \times n}, \tag{26}$$

where $R$ is the discretized Radon transfrom [7] that measures the projection images from different angles. Note that for CT applications, $R^T$ corresponds to performing backprojection (BP), and $R^\dagger$ corresponds to performing filtered backprojection (FBP). We choose

$$W = R^\dagger, \quad A = I - R^T(RR^T)^\dagger R, \quad b_i = R^T(RR^T)^\dagger y_i, \quad y_i \sim q(y_i|y), \tag{27}$$

where the choice of $A$ reflects that the Radon transform is not orthogonal, and we need the term $(RR^T)^\dagger$ as a term analogous to the filtering step. Indeed, this form of update is known as the algebraic reconstruction technique (ART), a classic technique in the context of CT [16]. We note that this choice is different from what was proposed in [40], where the authors repeatedly apply projection/FBP by explicitly replacing the sinogram in the measured locations. From our experiments, we find that repeated application of FBP is highly numerically unstable, often leading to overflow. This is especially the case when we have limited resources for training data (we use ~4k, whereas [40] uses ~50k), as we further show in Section 5.

Algorithm for SV-CT reconstruction uses PC sampler (VE-SDE), where we use MCG step after one sweep of corrector-predictor update. We note that this is a design choice, and one may as well use the MCG update step after both the predictor and corrector steps, as was proposed in [41]. We set $N = 2000, \alpha = 0.1/\|R^\dagger(y - R\hat{x}_0)\|$.

## D   Generative process of the proposed method

In Fig. 7, we depict the comparison of the generative process between the two methods: score-SDE [41], which relies on alternating projections; and our method, which utilizes MCG as correcting steps. In Fig. 7 (a), we can clearly see the unnatural boundary between the masked and the unmasked region forming, and evolving as $t \to 0$, without getting corrected (Visible more clearly in $\hat{x}_0$). On the other hand, thanks to the additional gradient step that *corrects* the errors in the boundary, we see a much more natural evolution of the signal as $t \to 0$ in Fig. 7 (b).

# E Limitations

There exists a limitation specifically for the ImageNet dataset when using the proposed algorithm for inpainting. Specifically, as shown in Fig. 8, for the case of half-mask (i.e. the left or right half of the image is zeroed-out), we often see the reconstructions are generated showing symmetries that are unrealistic. Note that this kind of effect is not observed in our FFHQ experiments. Hence, we conjecture that this phenomenon arises from the imperfectness of the learned score function $s_\theta$. Namely, due to the ImageNet dataset being much more diverse and therefore widely known to be a much harder dataset to learn, the subobtimality of the score function may be greater than the FFHQ score function. This could possibly lead to such deficiencies.

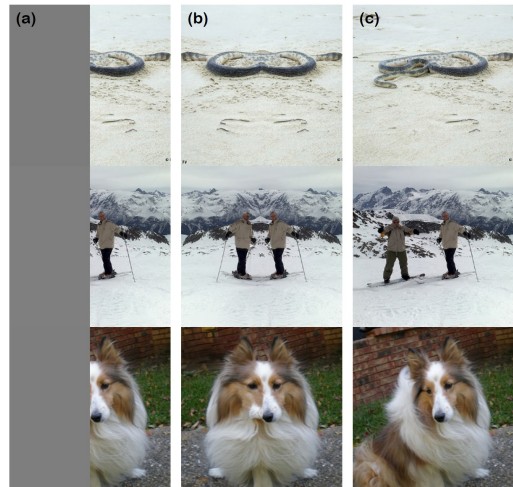

Figure 8: Limitations of the proposed method. (a) Measurement, (b) reconstruction with the proposed method, (c) ground truth.

# F Experimental Details

## F.1 Implementation details

**Training of the score function** For inpainting experiments, we take the pre-trained score functions that are available online (FFHQ[4], imagenet[5]). For CT reconstruction experiment, we train a `ncsnpp` model with default configurations as guided in [41] with the VE-SDE framework. The model was trained for 200 epochs with the full training dataset, with a single RTX 3090 GPU. Training took about one week wall-clock time.

**Required compute time for inference** All our sampling steps detailed in Algorithm C was performed with a single RTX 3090 GPU. The inpainting algorithm based on ADM [13] takes about 90 seconds (1000 NFE) to reconstruct a single image of size 256×256. Our colorization and CT reconstruction algorithm based on score-SDE [41] takes about 600 seconds (4000 NFE) to infer a single 256×256 image.

**Code Availability** We will open-source our code used in our experiments upon publication to boost reproducibility.

## F.2 Comparison methods

### F.2.1 Inpainting, Colorization

**Score-SDE** Score-SDE [41] demonstrated that unconditional diffusion models can be adopted to various inverse problems, such as inpainting and colorization. Our method without the MCG step is identical to score-SDE, and hence we use the same score function, parameters, and sampler as used in the proposed method for reconstruction.

**RePAINT** RePAINT [32] proposes to iterate between denoising-noising steps multiple times in order to better incorporate inter-dependency between the known and the unknown regions in the case of image inpainting. We use the same score function and sampler for RePAINT as in the proposed method. Following the default configurations in [32], we take $N = 200$ (corresponding to $T$ in [32]), and $U = 10$, where $U$ denotes the count of iterated denoising-noising steps used within a single update index $i$.

---

[4] https://github.com/jychoi118/ilvr_adm
[5] https://github.com/openai/guided-diffusion

**DDRM** DDRM [25] demonstrates that linear inverse problems can be solved via diffusion models by decomposing the generative process with singular value decomposition (SVD), and performing reverse diffusion sampling in the spectral space. The same score function adopted for the proposed method is used. Using the notations from [25], we choose $\sigma_{\boldsymbol{y}} = 0$, as we are aiming to solve noiseless inverse problem, and $\eta = 0.85, \eta_b = 1$. The number of NFE is set to 20 with the DDRM sampling steps.

**LaMa** LaMA contains fast Fourier convolution in generator architecture for reconstructing images. We trained the model from scratch using adversarial loss with r1 regularization term with its coefficient 10 and gradient penalty coefficient 0.001. Adam optimizer is used with the fixed learning rate of 0.001 and 0.0001 for discriminator network. For FFHQ and Imagenet dataset, 500k iterations of trainings were done with batch size of 8.

**AOT-GAN** AOT-GAN consists of a deep image generator with a AOT block which consists of multiple length of residual blocks in parallel. The discriminator is the same architecture with PatchGAN from [55]. We trained the model from the scratch with 0.0001 learning rate using Adam optimizer $\beta_1 = 0$ and $\beta_2 = 0.9$ for both FFHQ and Imagent dataset. 500k iterations of trainings were done with batch size of 8. Also, for style loss and the perceptual loss, VGG19 [39] pretrained on ImageNet [12] was used.

**ICT** Image completion transformer (ICT) consists of two modules - a transformer model that follows the tokenization procedure to process information in the lower dimensional space, and another guided upsampling module to retrieve the data dimensionality. The encoded features are sampled from a probability distribution via Gibbs sampling, such that one can capture multimodal reconstructions from the same measurement. For both the FFHQ and Imagenet dataset, we used pretrained models provided by the authors.

**IAGAN** Image adaptive GAN (IAGAN) uses a pre-trained generator and adapts it at test time for the given forward model. Specifically, following compressed sensing using generative model (CSGM) [4], one initializes the latent vector $\boldsymbol{z}$ such that $\boldsymbol{z}^* = \arg\min_{\boldsymbol{z}} \|y - AG_\theta(\boldsymbol{z})\|$. Then, the latent code and the neural network parameters are jointly optimized through some iterations of $\boldsymbol{z}^{**}, \theta^* = \arg\min_{\boldsymbol{z},\theta} \|y - AG_\theta(\boldsymbol{z})\|$. The final result is achieved by the forward pass through the generator, after which follows the projection into the measurement subspace. For tuning the generator, we follow the default configurations from the official codebase. Since the codebase uses a GAN that generates 1024×1024 images, we downscale the result into 256×256 image as a final post-processing step.

**DSI** DSI is structured with the combination of VQ-VAE [46], structure generator and texture generator. The architectures were trained separately, with Adam optimizer. When inference, only structure and texture generator was used. We trained the model from scratch. During optimization, the structure generator used linear warm-up schedular and square-root decay schedule used in [36]. We used Adam optimizer on training all models with learning rate of 0.0001 and $\beta_1 = 0.5$ using exponetial moving average (EMA). Training was done for 500k iteration for both FFHQ and Imagenet dataset.

**cINN** cINN is an invertible neural network which can take in additional conditions as input, and in our case grayscale images. We train the model using default configurations as advised in `https://github.com/VLL-HD/conditional_INNs` without modifications. FFHQ model was trained with the learning rate of 0.0001 for 100 epohcs using the Adam optimizer. LSUN bedroom model was trained with the learning rate of 0.0001 for 30 epochs.

**pix2pix** Pix2pix is a variant of conditional GAN (cGAN) that takes in as input, the corrupted image. The model is trained in a supervised fashion, with the loss consisting of the reconstruction loss, and the adversarial loss. As the discriminator architecture, we adopt patchGAN [21], and utilize the LSGAN [33] loss, weighting the adversarial loss by the value of 0.1. Similar to cINN, FFHQ model was trained with the learning rate of 0.0001 for 100 epochs using Adam optimizer. LSUN bedroom model was trained with the same configuration for 30 epochs.

### F.3 CT reconstruction

**Score-CT** We use the hyper-parameters as advised in [40] and set $\eta = 0.246, \lambda = 0.841$. The measurement consistency step is imposed after every corrector-predictor sweep as in the proposed method.

**SIN-4c-PRN** Directly using the official implementation[6] [50], we train the sinogram inpainting network (SIN) with the AAPM dataset for 200 epochs with the batch size of 8, and learning rate of 0.0001. We train two models separately for different number of views - 18, and 30.

**cGAN** We adopt the implementation of cGAN [15] from SIN-4c-PRN repository [6]. We train the two separate networks for 18 view, and 30 view projection, with the same configuration - 200 epochs, learning rate of 0.0001, and batch size of 8.

**FISTA-TV** We perform FISTA-TV [3] reconstruction using `TomoBAR` [45], together with the `CCPi` regularization toolkit [27]. Leveraging the default setting, we use the least-squares (LS) data model, and run the FISTA iteration for 300 iterations per image, with the total variation regularization strength set to 0.001.

## G Further Experimental Results

We provide extensive set of comparison study for each task in Fig. 9, 11, and 12. Furthermore, in order to illustrate the ability of our method to generate multimodal reconstructions given a measurement, we present further experimental results of inpainting and colorization in the following figures: Fig. 13, 14, 15, and 16

---

[6]`https://github.com/anonyr7/Sinogram-Inpainting`

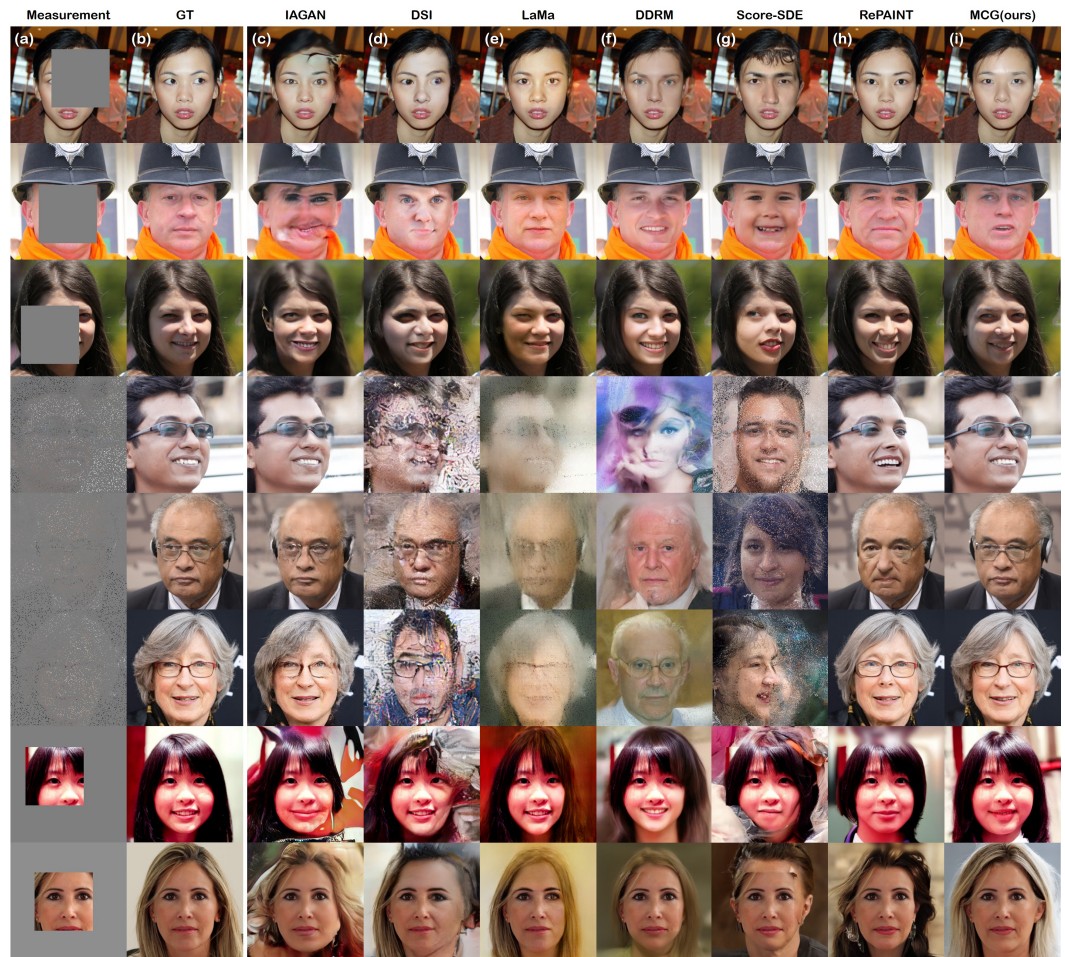

Figure 9: Inpainting results on FFHQ 256×256 data. (a) Measurement, (b) ground truth, (c) IAGAN [20], (d) DSI [35], (e) LaMa [43], (f) DDRM [25], (g) score-SDE [41], (h) RePAINT [32], (i) MCG (ours).

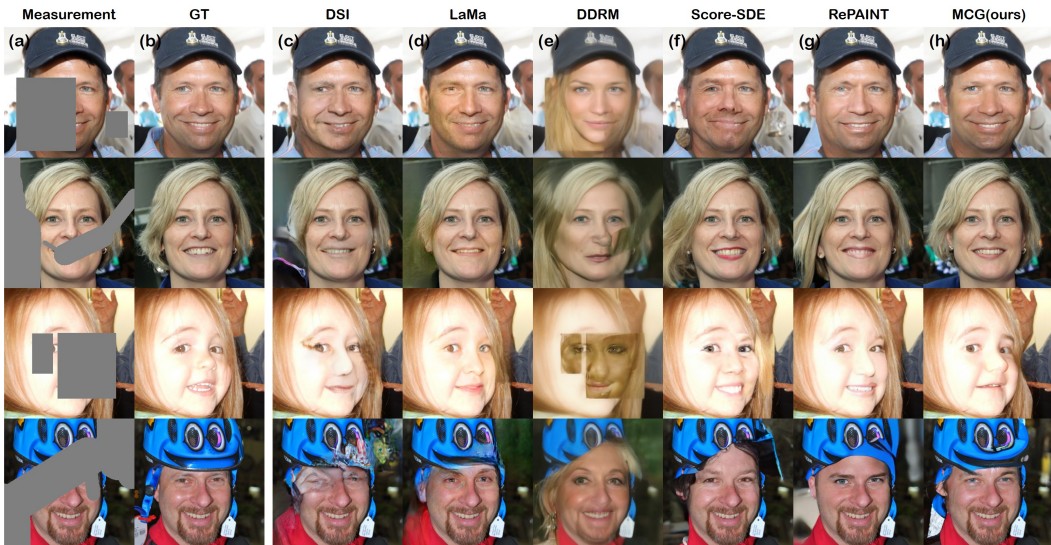

Figure 10: Inpainting results on FFHQ 256×256 data with the LaMa [43] wide mask. (a) Measurement, (b) ground truth, (c) DSI [35], (d) LaMa [43], (e) DDRM [25], (f) score-SDE [41], (g) RePAINT [32], (h) MCG (ours).

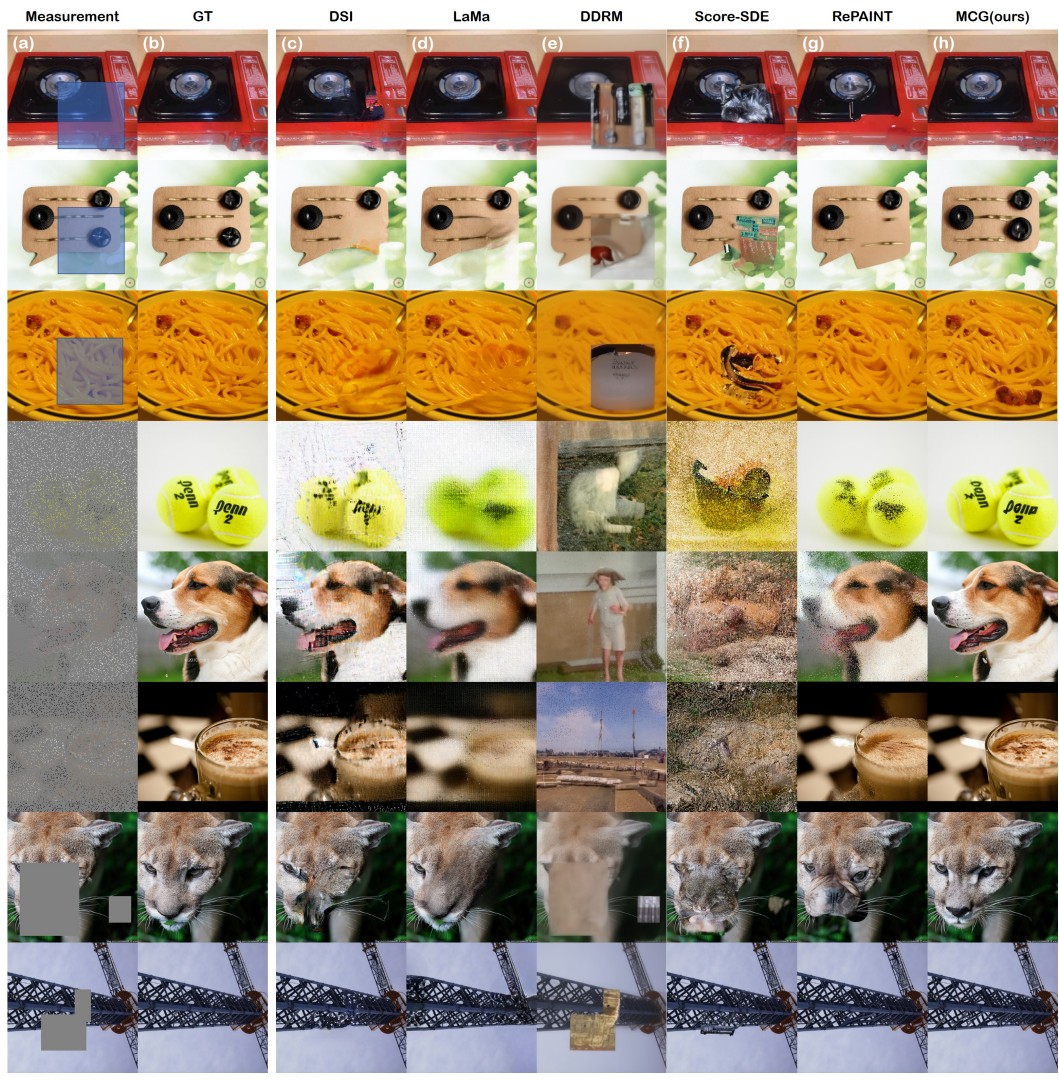

Figure 11: Inpainting results on ImageNet 256×256 data.(a) Measurement, (b) ground truth, (c) DSI [35], (d) LaMa [43], (e) DDRM [25], (f) score-SDE [41], (g) RePAINT [32], (h) MCG (ours).

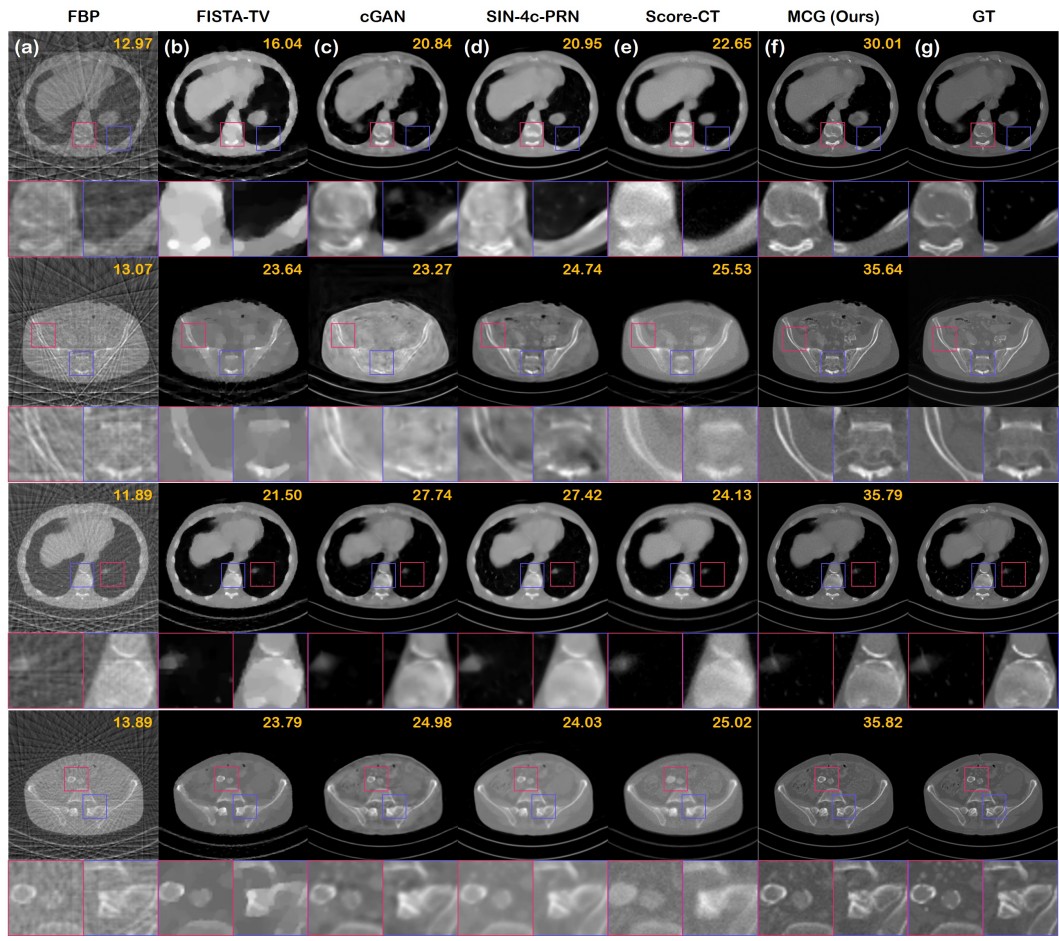

Figure 12: Sparse view CT reconstruction results on AAPM 256×256 data.(a) FBP, (b) FISTA-TV [3], (c) cGAN [15], (d) SIN-4c-PRN [50], (e) Score-CT [41], (f) MCG (Ours), (g) ground truth (GT).

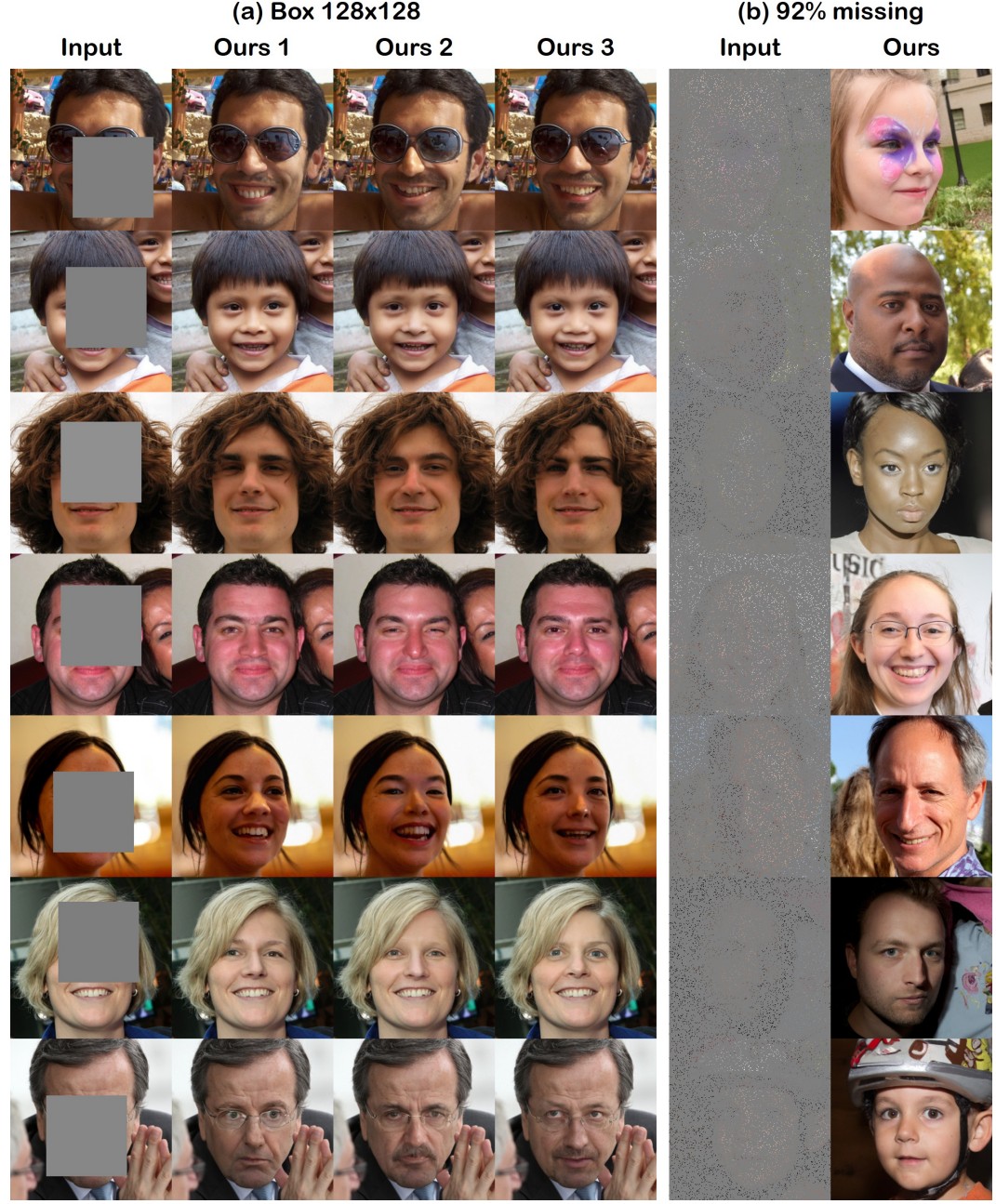

Figure 13: Inpainting results on FFHQ 256×256 data with MCG. (a) Inpainting of 128×128 box region. We show three stochastic samples generated with the proposed method. (b) 92 % pixel missing imputation.

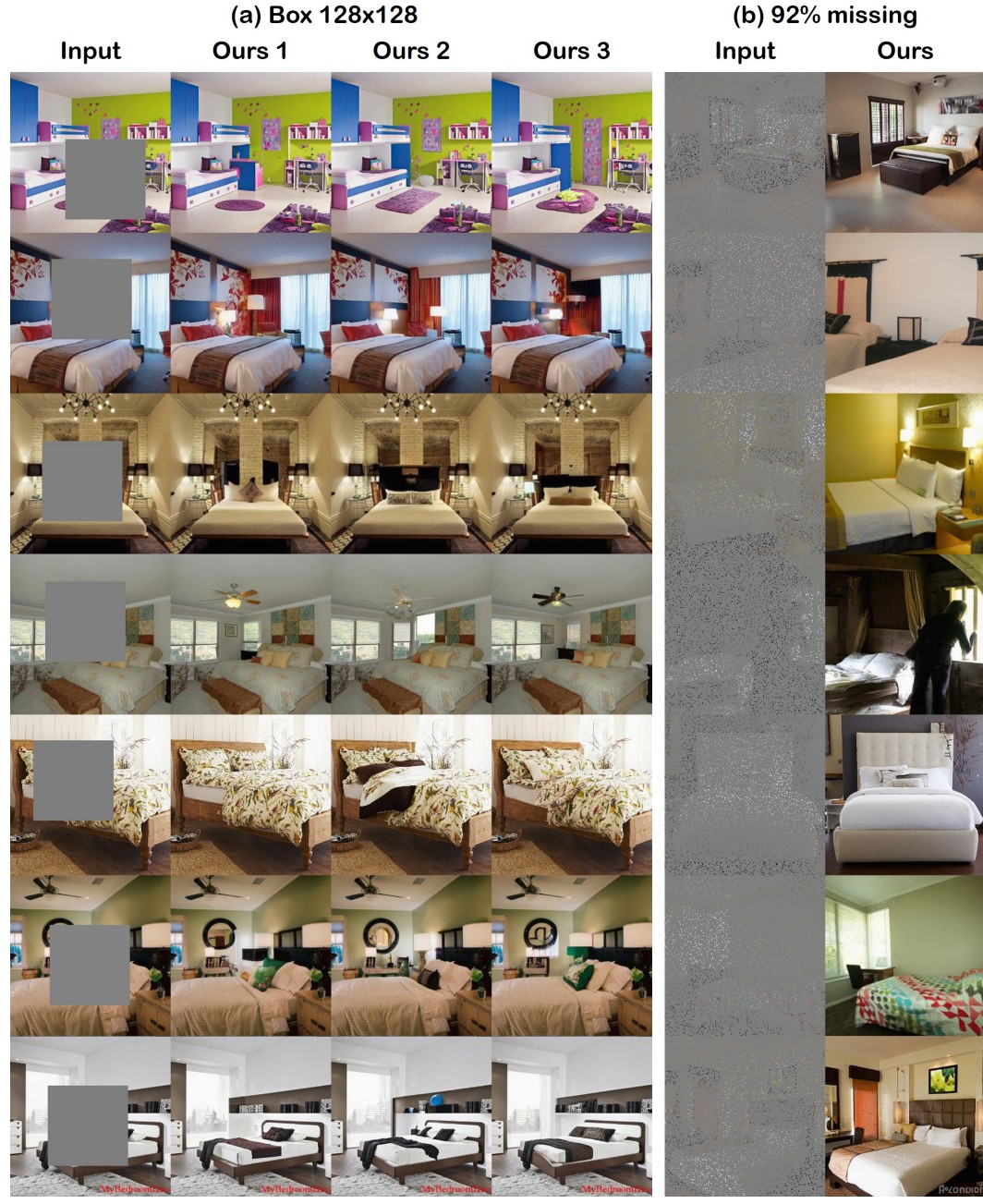

Figure 14: Inpainting results on LSUN-bedroom $256\times256$ data with MCG. (a) Inpainting of $128\times128$ box region. We show three stochastic samples generated with the proposed method. (b) 92 % pixel missing imputation.

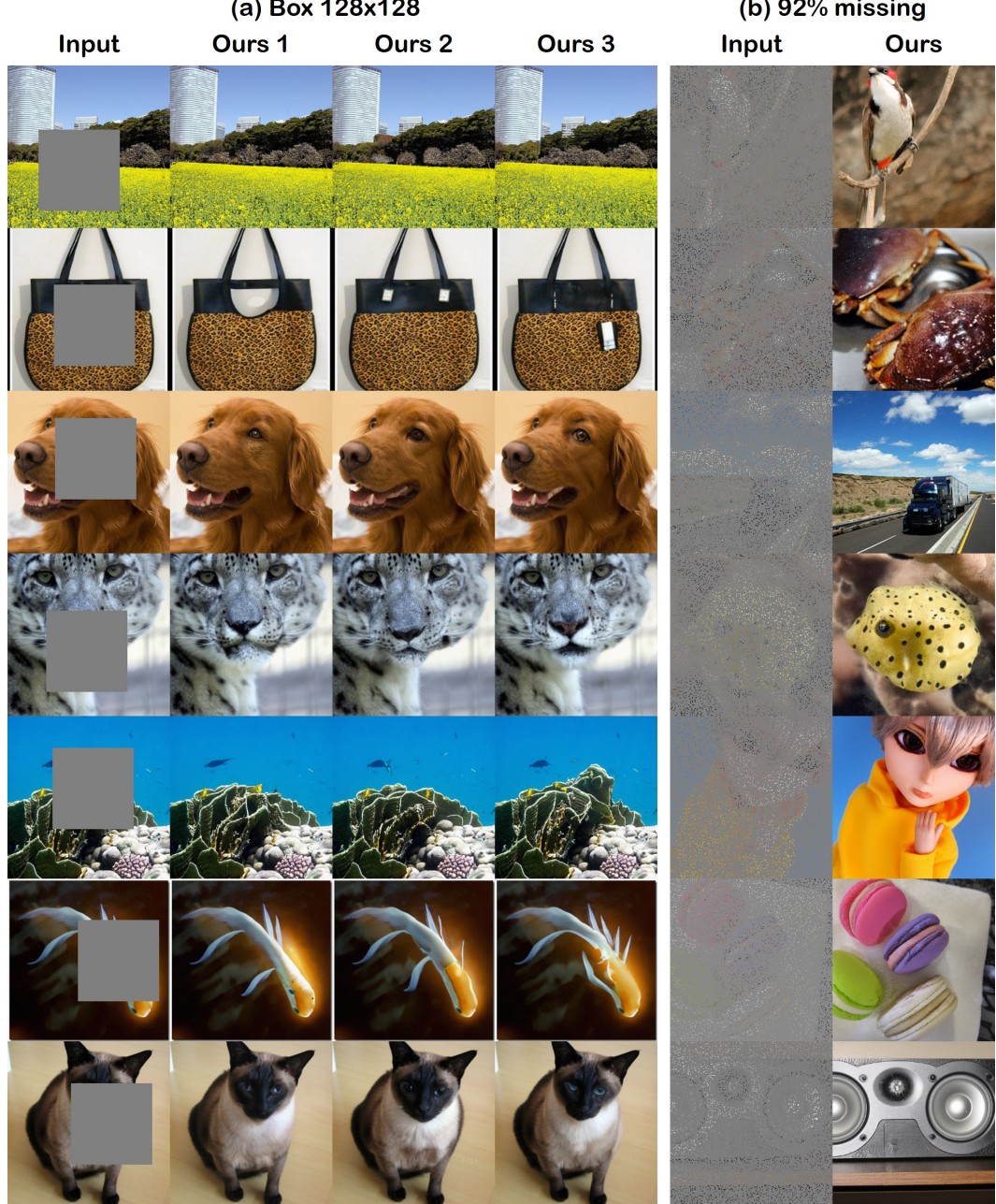

Figure 15: Inpainting results on ImageNet 256×256 data with MCG. (a) Inpainting of 128×128 box region. We show three stochastic samples generated with the proposed method. (b) 92 % pixel missing imputation.

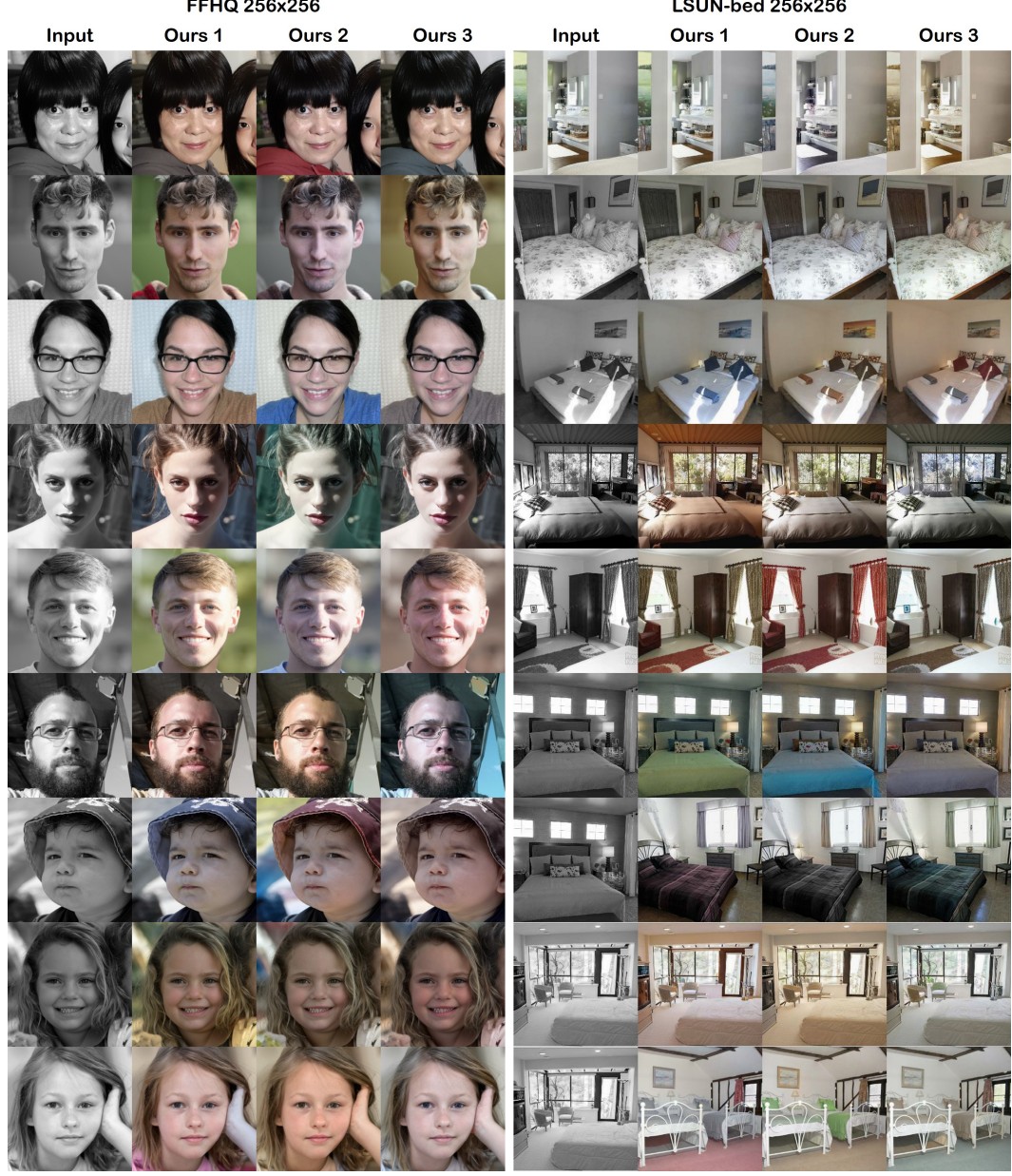

Figure 16: Colorization results on (left) FFHQ 256×256 dataset, and (right) LSUN-bedroom 256×256 dataset. We show 3 different reconstructions for each measurement that are sampled with the proposed method.