# OpenReview forum: "Improving Diffusion Models for Inverse Problems using Manifold Constraints"
_NeurIPS.cc/2022/Conference — NeurIPS 2022 Accept_

### Official Review · Reviewer_xYeQ · 2022-07-08

**Rating:** 6
**Confidence:** 4
**Soundness:** 2 fair
**Presentation:** 3 good
**Contribution:** 2 fair

**Summary:**

The paper suggests to improve the performance a diffusion-based reconstruction scheme by adding to its updates the gradient of a log-likelihood term that uses the link between $x_i$ to $x_0$.
Some mathematical motivation is provided.

**Questions:**

My questions and comments are listed above.

**Limitations:**

My questions and comments are listed above.

**Strengths And Weaknesses:**

(1). The introduction part is lacking. You should mention other methods that use the same pre-trained priors for solving many (almost arbitrary) reconstruction tasks (i.e., in an unsupervised fashion).
For example, GAN-based reconstruction methods [a],[b] and the powerful plug-and-play (P&P) denoisers approach [c],[d],[e].
The P&P denoisers approach [c], which is much older than diffusion-based reconstruction methods, commonly uses general-porpose CNN denoisers to impose the prior within iterative schemes [d], [e].
So, in fact, diffusion-based reconstruction methods can be understood as P&P variant with class-specific denoisers (determined by the training data) that can handle extreme noise levels (which makes them generative models).
I expect these topics to be discussed in the introduction.

(2). The continuous formulation of the diffusion process in the introduction part seems redundant.

(3). Fix the notation: in Eq. 15 you mean y rather than y0, right?
Also, the "argument" of $\hat{x}_0(\cdot)$ is $x_i$, right? this should be clearly written.

(4). The presentation of the MCG ingredient needs to be improved. Essentially, the "manifold constraint" is the information on the prior of the signal: the transformation from $x_i$ to $x_0$ (e.g.,the constraint in Eq. 13 is independent of $y$).
Yet, in the term that is added to Eq. 15 (compared to Eq. 7) you already connect it with the log-likelihood term (data fidelity term).
Thus, it should be clearly stated throughout the paper that this new term combines data fidelity term and the prior information of the transformation from $x_i$ to $x_0$.
Furthermore, I tend to believe that there exist other works that also use (perhaps in a similar way) the log-likelihood function in their iterative diffusion reconstruction schemes. Please make sure that you cover the exiting literature.

(5). Note that for $\alpha=0$ your additional term is canceled. Thus, your experiments should include ablation study and also some discussion and examination of the effect of different values of alpha.

(6). Please verify that there is no mismatch between Eq. 15 and the algorithms that you actually use in the experiments.
For example, in Algorithm 1 in Appendix C, line 10 is not aligned with Eq. 15 (alpha is multiplied with a matrix).
Moreover, observe that your MCG component in line 10 of Algorithm 1 vanishes(!) when you compute (I-P'P) * d/dxi ||P'*(y0-P*x0)||:

(I-P'P) * d/dxi ||P'*(y0-P*x0)|| = (I-P'P) * d/dx0 ||P'*(y0-P*x0)|| * dx0/dxi = (I-P'P) * P'P*(P*x0-y0) * dx0/dxi = 0,

because (I-P'P) * P'P=0, since for inpainting P is m rows of the identity matrix I_n so P'P is a projection matrix.

(7). In your algorithm, *all* the choices of $A$ and $b$ for the different tasks (including CT) makes Eq. 16 (and Eq. 8) coincide with the "back-projection step" in [e].
The iterative usage of such a data consistency step with pre-trained denoisers (which, as mentioned above, is very similar to diffusion-based reconstruction) has been proposed in [e] (and theoretically analyzed in follow-up works). Such very related works should be mentioned.

(8). The competitors in the experiments seem somewhat weak. For example, the recovery that is presented to some of them changes even the known pixels in inpainting. Recoveries that do not utilize such known information are obviously weak. Hence, it would have been interesting and informative to see comparison with strong (non-naive) GAN-based reconstruction methods, such as [b] (obviously, with GANs that are trained on the same training data as the other methods, e.g., by using common datasets and GANs).
The method in [b] is a clear example of a method that does not fall into the authors' statement that "the ability to maintain perfect measurement consistency ... is often not satisfied for unsupervised GAN-based solutions".

(9). In the experiments section, you must include a discussion on the number of iterations and run-time of the different methods that are examined.


[a] Bora, A., Jalal, A., Price, E. and Dimakis, A.G., 2017, July. Compressed sensing using generative models. In International Conference on Machine Learning (pp. 537-546). PMLR.

[b] Hussein, S.A., Tirer, T. and Giryes, R., 2020, April. Image-adaptive GAN based reconstruction. In Proceedings of the AAAI Conference on Artificial Intelligence (Vol. 34, No. 04, pp. 3121-3129).

[c] Venkatakrishnan, S.V., Bouman, C.A. and Wohlberg, B., 2013, December. Plug-and-play priors for model based reconstruction. In 2013 IEEE Global Conference on Signal and Information Processing (pp. 945-948). IEEE.

[d] Zhang, K., Zuo, W., Gu, S. and Zhang, L., 2017. Learning deep CNN denoiser prior for image restoration. In Proceedings of the IEEE conference on computer vision and pattern recognition (pp. 3929-3938).

[e] Tirer, T. and Giryes, R., 2018. Image restoration by iterative denoising and backward projections. IEEE Transactions on Image Processing, 28(3), pp.1220-1234.

---

> ### Author Response · Authors · 2022-08-01
> **Thank you for your questions and comments**
>
> Thank you for your constructive comments and thorough review. We would like to address your concerns and questions below:
>
> **Q1. Lack of relevant works: Plug-and-play(PnP), GAN, etc.**
>
> A. Thank you for pointing out the relevant works. Yes, diffusion models are very related to PnPs, and can be seen as generative variants of such models. As we focus only on the current problems of diffusion model-based inverse problem solvers and how we aim to solve such problems, we have included the discussion about both the PnP-based approach, and the GAN-based approach in Section 2. Related works.
>
> **Q2. Redundant formulation of continuous diffusion process**
>
> A. Agreed. We had to maintain the key formulations (e.g. forward/reverse SDEs) in the section but trimmed some of the redundant details.
>
> **Q3. Fix notation of eq.(15)**
>
> A. Fixed.
>
> **Q4. Presentation of the MCG ingredient**
>
> A. This is a very good point. We have modified our explanation about our MCG term. Specifically, we have now revised the main text in a way that follows the standard Bayes rule, followed by the additional Manifold Constraint and the data consistency.
>
> Per your second question, to the best of our knowledge, there exists only a single work [1] that directly tries to use the gradient of the log-likelihood to solve inverse problems. We have included the discussion in section 2.2.
>
> **Q5. Ablation study about the effect of varying values of $\alpha$**
>
> A. Please refer to the ablation studies paragraph that was added to the results section. Here, we show the variation in the performance as we change the value of $\alpha$.
>
> **Q6. Mismatch btw. eq (15), and the algorithm 1**
>
> A. Thanks for the careful reading and pointing out the typo. We would like to kindly correct the matrix calculus by $\frac{\partial}{\partial \mathbf{x}_i}\|\| P^T (\mathbf{y}_0 - P\mathbf{x}_0) \|\|_2^2 = \frac{\partial}{\partial \mathbf{x}_i} [(\mathbf{y}_0 - P\mathbf{x}_0)^T P P^T (\mathbf{y}_0 - P\mathbf{x}_0)] = 2\frac{\partial \mathbf{x}_0}{\partial \mathbf{x}_i}^T P^T P P^T (\mathbf{y}_0 - P \mathbf{x}_0)$.
>
> Especially, this computation is the key step in the proof of theorem 1. In the proof, the projection matrix $\frac{\partial \mathbf{x}_0}{\partial \mathbf{x}_i}^T$ (by proposition 2) transforms the vector $P^T P P^T (\mathbf{y}_0 - P\mathbf{x}_0)$ aligning it parallel to the data manifold. We have corrected algorithm 1 such that it matches eq. (15), and the code implementation. $\alpha$ should not be multiplied by the matrix.
>
> **Q7. Related work that uses back-projection steps with denoising**
>
> A. Thanks for making us aware of the relevant work. We have included the reference and the discussion.
>
> **Q8. Comparison methods are weak. comparison with IAGAN [2]**
>
> A. Comparison with IAGAN [2] was included for the FFHQ inpainting task, which is indeed a strong competitor. Also, we were unaware of GAN-based methods that can maintain perfect measurement consistency such as [2], and hence we tempered the statement in our manuscript. We would also like to emphasize those methods such as LaMa [3], and RePAINT [4] are the current state-of-the-art methods in their experimental settings (i.e. using LaMa wide, and thin masks). The only reason that they might seem weak is due to the extremely hard problem setting that we propose.
>
> **Q9. Discussion of number of iterations and time**
>
> A. For diffusion models, the major bottleneck that consumes most of the running time is the forward pass through the approximated score functions (i.e. denoising steps). We have included an ablation study in the results section comparing the different diffusion model-based methods and showed the difference in the performance in each method as we vary the NFE.
>
> **References**
>
> [1] Kawar, Bahjat, et al. "Denoising diffusion restoration models." arXiv preprint arXiv:2201.11793 (2022).
>
> [2] Hussein, Shady Abu, Tom Tirer, and Raja Giryes. "Image-adaptive GAN based reconstruction." AAAI 2020
>
> [3] Suvorov, Roman, et al. "Resolution-robust large mask inpainting with fourier convolutions." WACV 2022.
>
> [4] Lugmayr, Andreas, et al. "Repaint: Inpainting using denoising diffusion probabilistic models." CVPR. 2022.

---

> ### Author Response · Authors · 2022-08-07
> **Follow up before the discussion period is closed.**
>
> Thank you again for your review and comments. As we are approaching the end of the discussion period, we would like to ask the reviewer if our review and response have properly addressed the initial concerns. We are more than happy to discuss and clarify if you have further comments.

---

> > ### Comment · Reviewer_xYeQ · 2022-08-07
> > **Response**
> >
> > I think that the modifications improve the paper. Though, I still have the following comments.
> >
> >
> > Regarding Q4, as I mention before, it seems very unlikely that "there exists only a single work [1] that directly tries to use the gradient of the log-likelihood to solve inverse problems" as you claim.
> > For example, please check [a] (its references and citations). Note that [a] is named "plug-and-play" but it does sampling similar to what is done with score/diffusion-based reconstruction methods. Again, this follows from the similarity that is mentioned above.
> > Combining log-likelihood + (approximate) log-prior (to approximate posterior) and differentiating them for optimization/sampling is a very common estimation approach.
> >
> >
> > Regarding Q9, I would like to see runtime comparison of the methods (with similar HW setting) that will allow comparing them to non-diffusion based methods, e.g., IAGAN (state its runtime as well), that use much simpler models.
> >
> >
> > The ablation study improves the paper. Yet, I think that a very important examination should be added (related to Q4):
> > The case where you add the negative-log-likelihood term but without the transformation from x_i to x_0, i.e., without using the manifold knowledge (that is assumed to be correctly captured by the network).
> > In this case, use y_{i} or y_{i+1} (rather than y) that is sampled from y to match the noise level of x_i, and weigh the log-likelihood term according the the noise level.
> >
> >
> > [a] Laumont, R., Bortoli, V.D., Almansa, A., Delon, J., Durmus, A. and Pereyra, M., 2022. Bayesian imaging using Plug & Play priors: when Langevin meets Tweedie. SIAM Journal on Imaging Sciences, 15(2), pp.701-737.

---

> > > ### Author Response · Authors · 2022-08-07
> > > **Thank you for your constructive comments.**
> > >
> > > We would like to greatly thank the reviewer for involving in the discussion and providing us with some important feedback. Below, we further address your concerns:
> > >
> > > **Q1.Related works on posterior sampling (PnP-type)**
> > >
> > > We would like to clarify our comment that “there exists only a single work that directly tries to use the gradient of the log-likelihood to solve inverse problems”. While we were aware that there exists a vast literature on combining log-likelihood and log-prior to achieve posterior sampling, we were restricting ourselves to *diffusion model-based* inverse problem solvers specifically that directly try to estimate the log-likelihood.
> > >
> > > Having said that, [1] is indeed very related to the proposed method and is definitely worth discussing in the paper. Together with some of the works in the context of PnP models, we have added a paragraph in the end of chapter 3, which extends our discussion on related works that aim for posterior sampling leveraging pre-trained denoisers.
> > >
> > > **Q2. Runtime evaluation**
> > >
> > > We agree that including specific runtimes for comparison methods would be of benefit. Please see table 4 which contains the runtime measured with a commodity GPU. In order to make our comparison fair, we matched the image size also for IAGAN, and run the optimization algorithm. Please note that the current runtime for IAGAN is measured as implemented in the official codebase, which may be further accelerated via parallelization of the optimization procedure, as stated in this [code](https://github.com/shadyabh/IAGAN/blob/19311198f421ce49aeb9caa54eb4b69efa461bcd/IAGAN.py#L111). We will measure the run-time of IAGAN when the parallelization takes place, and replace the current runtime in Table 4 with the newly measured values in our camera-ready version. We are also including table 4 here for the reviewer’s convenience.
> > >
> > > | Method    | Wall clock time [s] |
> > > |-----------|---------------------|
> > > | Score-SDE | 38.68               |
> > > | RePAINT   | 247.6               |
> > > | DDRM      | 2.117               |
> > > | LaMa      | 0.629               |
> > > | AOT-GAN   | 0.082               |
> > > | ICT       | 144.6               |
> > > | DSI       | 36.64               |
> > > | IAGAN     | 518.47              |
> > > | **Ours**  | 81.59               |
> > >
> > > **Q3. Further ablation study by using non-denoised gradient steps**
> > >
> > > Thank you for your constructive comment. We have now extended our ablation study to contain your suggestions. When taking the gradient step, we implemented the gradient of the log-likelihood term without transforming $\mathbf{x}_i$ to $\mathbf{x}_0$, and sampled $\mathbf{y}_i$ to match the noise level of $\mathbf{x}_i$. From the experiments, we show that taking such an approach heavily degrades the performance, especially when not used together with projections. This indeed shows that the proposed Tweedie’s denoising is the key to producing superior results. Please refer to the modified Fig. 5(a) and Section 5. Ablation studies paragraph for details. We are also including the modified table for the reviewer’s convenience.
> > >
> > > | Method       | LPIPS($\downarrow$) | MSE(MC) |
> > > |--------------|-------|---------|
> > > | Proj.        | 0.138 | 0       |
> > > | MCG          | 0.124 | 10.7    |
> > > | Grad         | 0.271 | 12.9    |
> > > | Grad + Proj. | 0.128 | 0       |
> > > | **Ours**         | 0.089 | 0       |
> > >
> > > **References**
> > >
> > > [1] Laumont, Rémi, et al. "Bayesian imaging using Plug & Play priors: when Langevin meets Tweedie." SIAM Journal on Imaging Sciences 15.2 (2022): 701-737.

---

### Official Review · Reviewer_ze1d · 2022-07-08

**Rating:** 6
**Confidence:** 4
**Soundness:** 3 good
**Presentation:** 3 good
**Contribution:** 3 good

**Summary:**

In this paper, the authors propose a new methodology to perform conditional sampling using denoising diffusion models. Namely, they combine existing conditional score-matching techniques using a stochastic contraction with a slight change of the score. Indeed, instead of considering the unconditional score functional they consider the score function associated with the conditional distribution given by some manifold constraint. The authors show that this term is in fact complementary to the existing unconditional score term as the score term only (locally) project on the data manifold, whereas the additional term moves locally on the tangent space of this manifold. They evaluate their methodologies against some baselines and show promising results for inpainting, colorization and CT-scan reconstruction.

**Questions:**

* The authors state that "Accordingly, the score function ∇x log pt (x) in (2) should be replaced by the conditional score $\nabla_x \log p_t(x|y)$. Unfortunately, this strictly restricts the generalization capability of the neural network since the conditional score should be retrained whenever the conditions change." This is not necessarily true as the score could be trained using some amortization w.r.t. y. In this case $y$ can be simply passed as another input to the diffusion model.

* I don't really understand how the authors derive (14) from (13) and (6). From what I see, the authors are simply plugging $\nabla \log p(y|\hat{x}_0)$ in the drift of the backward Markov chain. This makes a lot of sense, but I find the explanation provided by the authors to be quite convoluted.

* Assumption 1 is not the manifold assumption as understood in the literature. Usually the manifold assumption ensures that the data lives on a lower dimensional manifold but do not impose any local linear structure.  It is quite hard to believe that the data of any reasonable image distribution is linear locally. I understand that the authors need this assumption to make their results rigorous but this should be clearly stated.

* I really enjoyed reading about Proposition 2 and Theorem 1. I think that together with Figure 2 they bring a lot of insight about the method. However, the authors do not discuss at all the effect of the projection step $A x_i' + b$. I would have really appreciated an ablation study comparing the effects of these two terms which both introduce some information about the measurement.

* I am not an expert in this domain but I have to admit that I am a bit surprised by the FID scores obtained by the authors which seem to be dramatically better than the rest of the methods. Could the authors provide details about the experimental process and in particular how they train the different models? For example do you perform hyperparameters tuning.

* I think that the comparison with DDRM is a bit biased because DDRM only use 20 steps to recover the data whereas in the appendix the authors indicate that they use between 1000 and 2000 steps depending on the task. The trade-off between sampling quality and NFE is not investigated. Similarly, when comparing with RePaint the authors chose N = 200.

**Limitations:**

The limitations are discussed in the conclusion.
I found this discussion to be appropriate.

**Strengths And Weaknesses:**

STRENGTHS:

* The paper introduces a new methodology for inpainting which makes sense from a theoretical point of view, is easy to implement and seems to yield good results.

WEAKNESSES:

* I think that a deeper discussion with recent works such as [1] is missing. It is not clear from reading the main document what is the state of the art conditional sampling method using diffusion models (at least previous to this work).

* In terms of novelty it seems that the proposed correction was already introduced in [2]. As a consequence, the method proposed by the authors can be see as a generalization to arbitrary linear problems (as stated by the authors) and therefore has limited novely.

* With the stochastic contraction term and the gradient term proposed by the authors it is not clear at all if we are sampling from the right posterior distribution. I understand that the method yields good visual results but in fields like CT-reconstruction the visual quality should not be the only metrics as we also need a models which provide good uncertainty quantification. Such considerations are missing here. As a first question: can you characterize what is the invariant measure of the proposed diffusion model?

[1] Kawar et al. (2022) -- Denoising diffusion restoration models
[2] Ho et al. (2022) -- Video diffusion models

---

> ### Author Response · Authors · 2022-08-01
> **Thank you for your questions and comments.**
>
> We thank the reviewer for their thorough and constructive review. We address the concerns and questions below:
>
> **Q1. A deeper discussion on the recent works**
>
> A. Thank you for pointing this out. Indeed, [1] establishes the state-of-the-art especially for solving noisy inverse problems with unconditional diffusion models. We have included the discussion in the related works section.
>
> **Q2. Deviation from posterior sampling, invariant measure of the proposed sampling?**
>
> A. This is indeed an interesting question. We agree that our additional constraints (to enforce the generative path on the manifold and additional data consistency) could lead to images not being sampled from the *correct* posterior distribution. However, note that our method still enjoys perfect data consistency, and hence only generates *feasible* solutions to the given problem. We can interpret that our samples are generated from the approximate posterior distribution, which is additionally regularized with our manifold constraint. Moreover, exact posterior sampling is not possible in the first place, due to the imperfect modeling of $p(x)$ with the diffusion model. Having said that, we do think that it could be a very promising direction of research to find a way to perform exact posterior sampling with our method (that is, up to optimization + parameterization errors).
>
> **Q3. Amortization for the conditional diffusion**
>
> A. We are aware that conditional diffusion models that rely on amortization of the conditioning exist in the literature (e.g. [2,3]). Nonetheless, these models are still restrictive, in the sense that when the measurement condition changes, one would have to retrain the model according to the conditioning. In the simplest case, $\times$4 super-resolution model will not be capable of performing $\times$16 super-resolution; MRI reconstruction model that was trained with 2D variable density mask will perform very poorly on 1D uniform sampling mask. In contrast, using our method, we are free from such restrictions and can use the **same model** for **all the different tasks** given at hand.
>
> **Q4. Derivation of the MCG**
>
> A. Thanks for the constructive comments. You are right that our method is recovered by plugging in $p(\mathbf{y}|\hat{\mathbf{x}}_0)$ to the reverse diffusion process! To clarify this, we have now revised the main text in a way that follows the standard Bayes rule, followed by the additional Manifold Constraint and the data consistency.
>
> As conditioning the model on measurement is effectively implemented by projections, our motivation was to further guide our generative process to stay on the manifold, and therefore we introduce a new set of constraints orthogonal to the original conditioning method. In terms of theory, the MCG step may have an error in the practical implementation of the score matching, etc. Therefore, to complement the error, the additional projection step is necessary.
>
> **Q5. Strong manifold assumption**
>
> A. Thanks for the constructive comments. Per your comment, we have toned down the claim, and in the revised version, we modify the title of the assumption and remark that it is a strong version of the manifold assumption.
>
> **Q6. Comparing the effect of MCG and projection step**
>
> A. Thanks for the constructive comments. Please refer to the Ablation studies paragraph that was included in the result section. From the ablation study, we can conclude that only using the MCG step induces some measurement consistency error, and combining MCG with projection will induce perfect consistency while improving the performance even further. In terms of theory, the MCG step may have an error in the practical implementation of the score matching, etc. Therefore, to complement the error, the projection step is necessary as shown in the ablation study.
>
> **Q7. Details of the experimental setup**
>
> A. Yes, this is detailed in Appendix E.2. Comparison methods. For most of the methods that have open-source code available, we used pre-trained checkpoints that were available on Github. For image datasets that had to be trained from scratch, we tried to abide closely by the advised settings in the original works and did some hyperparameter tuning with grid search to train the models properly.
>
> For RePAINT, by the time when we were writing the manuscript, the official codebase was unavailable, and we had to resort to our own re-implementation (see code in our supplementary material), where we tried to abide closest to the algorithm introduced in the paper. A few days ago, the official code was released with several heuristics that were not explicitly mentioned in the paper, thereby inducing huge gains in performance. We re-ran all the experiments with RePAINT, with improved results. Note that **the proposed MCG method still outperforms RePAINT**, albeit using the official code.

---

> > ### Author Response · Authors · 2022-08-01
> > **Continuation of our response**
> >
> > **Q8. Biased experiments with DDRM**
> >
> > A. The choice of NFEs was taken from the default setting of each paper (Note that for RePaint, N = 200 corresponds to 2000 NFEs, due to the iteration steps they take). However, as you said, we agree that it would be beneficial to thoroughly compare diffusion-based methods by matching their NFEs to the same level. For that, we have included an ablation study by showing how each method performs with NFEs in the range [20, 1000]. We see that for low NFE regime (<= 50), DDRM excels. In contrast, for high NFE regime (>= 100), MCG excels. This is most likely due to the sampler that is utilized in each method: DDRM uses DDIM, while we use DDPM, and hence using advanced samplers to boost performance in low NFE regime would be prospective future research.
> >
> > **References**
> >
> > [1] Kawar, Bahjat, et al. "Denoising diffusion restoration models." arXiv preprint arXiv:2201.11793 (2022).
> >
> > [2] Saharia, Chitwan, et al. "Image super-resolution via iterative refinement." arXiv preprint arXiv:2104.07636 (2021).
> >
> > [3] Saharia, Chitwan, et al. "Palette: Image-to-image diffusion models." arXiv preprint arXiv:2111.05826 (2021).
> >
> > [4] Song, Yang, et al. "Score-based generative modeling through stochastic differential equations." ICLR 2021.
> >
> > [5] Chung, Hyungjin, Byeongsu Sim, and Jong Chul Ye. "Come-closer-diffuse-faster: Accelerating conditional diffusion models for inverse problems through stochastic contraction." CVPR 2022.

---

### Official Review · Reviewer_qC4w · 2022-07-10

**Rating:** 6
**Confidence:** 4
**Soundness:** 3 good
**Presentation:** 2 fair
**Contribution:** 3 good

**Summary:**

This paper proposes a framework for solving inverse problems using diffusion models as priors. The idea is to introduce another constraint in the formulation of the posterior distribution that enforces the generated sample to be on the manifold of natural images. The introduced constraint is motivated by the Tweedie formula, that relates the score function (of the noisy data distribution) and the best MMSE estimator (denoising).
Several experimental results on different imaging inverse problems show that the proposed method outperforms existing methods in FID, LPIPS, PSNR and SSIM metrics.

**Questions:**

For completeness, I re-list here some of the questions raised above:

1. Tweedie formula vs Noise2score.The key motivation seems to be coming from the Tweedie formula. With that in mind it would be better to introduce the Tweedie formula and relevant work better. See for example Section IIB in [Ong2019]:

[Ong2019] Ong, F., Milanfar, P. and Getreuer, P., 2019. Local kernels that approximate Bayesian regularization and proximal operators. IEEE Transactions on Image Processing, 28(6), pp.3007-3019.

2. Motivation for the additional constraint is not clear (Section 3.1). In particular Eq (14) deduction is unclear (\mathcal{C} is a set, so how should one interpret p(\mathcal{C} | x) ? Could you please elaborate more on how Eq (14) is introduced (motivation/deduction).

3. How is obtained W for each application, it is unclear where this W is coming from. Explaining better this term in (14) will help a lot to the understanding of the algorithm.

Other minor comments:
- l59 -  p_t(x) should be log p_t(x)
- Eq (3) - \epsilon is undefined maybe just add "where \epsilon <<1".
- Eq (18), Eq (20),  Eq (22). In all these equations it says "We chose W…" This implies that W is not deducted from the model but it's something that we need to choose. It is unclear how to do this and how it's related to the introduced motivation where everything seem to be coming naturally from the Tweedie formula.

I'm more than happy to increase my final rating if during the rebuttal we manage to clarify some of the aspects regarding this work.


**Limitations:**

Limitations are correctly discussed.

**Strengths And Weaknesses:**

Strengths:
- The paper introduces an interesting adaptation to traditional sampling methods motivated by the Tweedie formula.
- The paper is generally well written and the idea is well motivated.
- Several experimental results on different imaging inverse problems show that the proposed method outperforms existing methods in FID, LPIPS, PSNR and SSIM metrics.

Weakness:
- The paper exposition could be better:
- The paper relates the idea a lot to noise2score, but the key motivation is the Tweedie formula. With that in mind it would be better to introduce the Tweedie formula and relevant work better (see below)
- Motivation for the additional constraint is not clear (Section 3.1). In particular Eq (14) deduction is unclear (\mathcal{C} is a set, so how should one interpret p(\mathcal{C} | x) ?
- How is obtained W for each application.
- Theoretical findings should be better connected to the proposed algorithm.

In general I like the paper, the idea is interesting, and the experimental results show a clear win in performance. The main current issue to me is that the motivation and exposition is not clear enough. Most of the pieces are already there but a reorganization and better discussion is needed.

---

> ### Author Response · Authors · 2022-08-01
> **Thank you for your questions and comments.**
>
> We thank the reviewer for the valuable review. We address your concerns and questions below:
>
> **Q1. Tweedie's formula vs. Noise2Score, and the exposition**
>
> A. We agree that it would be more useful to formulate our idea using Tweedie’s formula rather than Noise2score directly. Now, section 2.3 is named “Tweedie’s formula for denoising”, and relevant discussions on works such as [1] were added.
>
> **Q2. Motivation for the additional constraint**
>
> A. To clarify this, we have now revised the main text in a way that follows standard Bayes, followed by the additional Manifold Constraint and the data consistency.
>
> **Q3. How $W$ are obtained for each application**
>
> A. $W$ is obtained for each application such that the gradient from the measurement domain targets the image domain. For inpainting, both domains remain in the image domain, hence our choice of $W=I$. For colorization and CT reconstruction, we take the pseudo inverse $W=H^\dagger$ (cf. For colorization the matrix is orthogonal and hence equal to the transpose), where $H$ is the measurement matrix. We note that the different choices of $W$ were selected for the best performance, but simply setting $W=I$ for all applications also suffices, still bringing large improvements from the previous methods. Hence, our method is robust to these choices.
>
> **References**
>
> [1] Ong, Frank, Peyman Milanfar, and Pascal Getreuer. "Local kernels that approximate Bayesian regularization and proximal operators." IEEE TIP 2019

---

> ### Author Response · Authors · 2022-08-07
> **Follow up before the discussion is closed.**
>
> Thank you again for your review and comments. As we are approaching the end of the discussion period, we would like to kindly ask the reviewer if our revision and response have clarified the initial concerns about the work. If you have further questions or comments, we would be more than willing to discuss and further clarify these aspects.

---

### Official Review · Reviewer_vg3b · 2022-07-12

**Rating:** 7
**Confidence:** 4
**Soundness:** 3 good
**Presentation:** 3 good
**Contribution:** 3 good

**Summary:**

The paper proposes a simple augmentation to existing score-based diffusion solvers for inverse problems by applying an extra correction term in the diffusion step. This term, in theory, captures the on-manifold component of the current estimation error.

-------------

After rebuttal: The authors have convinced me that the empirical experiments are sound, and have also tempered their theoretical claims. I am happy with the paper in its present state, as I am convinced that it presents a novel technique for inverse imaging, and achieves a solid improvement on the prior state of the art.

**Questions:**

See (Overview) in Strengths and Weaknesses for questions / points to address.

**Limitations:**

No.

**Strengths And Weaknesses:**

Strengths:
- The proposed step is elegant and computationally simple.
- The results, if correct (see below), suggest massive improvements in signal recovery.

Weaknesses:
- The main theorem does not state its assumptions clearly. Moreover, I don't believe they hold, and this is quite problematic for the proposed algorithm. (See overview.)
- The empirical experiments are somewhat problematic. (See overview.)
- (Minor) The use of Bayes rule in Section 3.1 to obtain Eqs. 11 and 12 is somewhat misleading; this is a general technique for conditional sampling in score-based diffusion models, see for e.g. [1, 2, 3].

Overview:
Overall, I am very excited by the direction proposed here, but I think there are currently major issues in the reasoning and execution of the paper. I believe the paper will be much stronger after addressing these problems. Ultimately, my concerns are two-fold:

Theoretically, I am not convinced that the main result of the paper, upon which the proposed algorithm rests, holds in practice. Theorem 1 requires Proposition 1 to hold, and Proposition 1 assumes that the score function is a *global minimizer* of the denoising loss (Eq. 9). This is nearly impossible in a diffusion model, since the denoising function at t=T (or t=0 in the reverse process) must "recover" an image from pure noise --- this is a highly ill-defined problem. Another way to see that this cannot hold: If the score function was a global minimizer, the generative process for unconditional diffusion models would take 1 inference step (in practice, it takes thousands of steps in the vanilla diffusion model). The suboptimality of the score function has consequences that extend past theory, as the central manifold constraint gradient (MCG) term also relies on the global optimality of the score function to define manifold projection (Eq. 13). Thus, this key assumption is quite central to the entire paper.

Empirically, the visual quality of competing work seems like it could be somewhat misrepresented, especially in image inpainting. For example, in Figure 3, the visual quality of RePaint [4] and LaMa [5] both differ greatly from that reported in their respective papers (see Figure 1 for each paper). Moreover, looking more closely at the experimental setup of the competing works RePAINT and LaMa, the two papers share the same experiments for inpainting, and establish a baseline. Namely, their inpainting task involves recovering an original image corrupted by "Wide" and "Narrow" masks. This paper uses a different set of masks, and reports significantly different LPIP scores for RePAINT and LaMa for this task. To be clear, I think it is fair to change the experimental setup. However, I wonder if the authors did not spend enough time tuning the competing models in this new setup, and consequently misrepresented the competing works.

[1] Dhariwal, P., & Nichol, A. (2021). Diffusion models beat gans on image synthesis. Advances in Neural Information Processing Systems, 34, 8780-8794.
[2] Sohl-Dickstein, J., Weiss, E., Maheswaranathan, N., & Ganguli, S. (2015, June). Deep unsupervised learning using nonequilibrium thermodynamics. In International Conference on Machine Learning (pp. 2256-2265). PMLR.
[3] Song, Y., Sohl-Dickstein, J., Kingma, D. P., Kumar, A., Ermon, S., & Poole, B. (2020). Score-based generative modeling through stochastic differential equations. arXiv preprint arXiv:2011.13456.
[4] Lugmayr, A., Danelljan, M., Romero, A., Yu, F., Timofte, R., & Van Gool, L. (2022). Repaint: Inpainting using denoising diffusion probabilistic models. In Proceedings of the IEEE/CVF Conference on Computer Vision and Pattern Recognition (pp. 11461-11471).
[5] Suvorov, R., Logacheva, E., Mashikhin, A., Remizova, A., Ashukha, A., Silvestrov, A., ... & Lempitsky, V. (2022). Resolution-robust large mask inpainting with fourier convolutions. In Proceedings of the IEEE/CVF Winter Conference on Applications of Computer Vision (pp. 2149-2159).

---

> ### Author Response · Authors · 2022-08-01
> **Thank you for the questions and the feedbacks**
>
> Thank you for your constructive and thoughtful review. Below, we would like to address your concerns and questions:
>
> **Q1. Strong assumptions in our theorem/propositions**
>
> A. Thanks for pointing out the typo in Eq. (3). The learned score function for our MCG step is from the denoising score matching cost function, which is now revised in (3).  We would like to kindly remind the reviewer that all the mathematical theory for the manifold constraint (Proposition 2 and Theorem 1) assumes that the learned score function is a global minimizer of the denoising score matching (3). Accordingly, as long as the neural network has sufficient representation power, our mathematical analysis holds.
>
> In practice, the global optimality of the learned score function becomes impractical, especially as $t$ approaches $T$. This can also be seen in Fig.F.1. of [5], where the error grows rapidly when $t > 0.6T$. On the other hand, the error norm stays very low when $t < 0.6T$, and thus we maintain that it is safe to say that the neural network closely approximates the ground truth score in the low-noise regime. Note that the erroneous high-noise regime is where the deviations from our theoretical analysis would arise. However, also note that this is a regime where the image just starts to form, with only the coarsest features visible in  $\hat{\mathbf{x}}_0$. Due to this reason, the gradient of the MCG term remains small in such regime, deviations from our theoretical analysis do not matter too much. In the important regime ($t < 0.6T$, low noise) where the norm of the gradient starts to grow and actually matter, our assumption approximately holds, which leads to increased performance in the reconstruction. Additionally, the successive application of the projection step in our method can complement the errors in MCG step as shown in the ablation study.
>
> Having said that, we agree that we have made rather strong assumptions in order to establish our theory. As stated in the general comment D, we have tempered our theoretical claims and clarified that our analysis builds on these strong assumptions.
>
> **Q2. Problems in the experiments**
>
> A. We would like to note that LaMa “Wide” and “Narrow” masks are the regimes where both LaMa[1] and RePAINT[2] do a great job, and even the “Wide” masks tend to be much easier to inpaint, compared to the two settings that we propose (box 128x128, and random 92%). For LaMa, these are the models that were trained with “Wide” and “Narrow” masks with full supervision, and it is natural that it will excel with the same settings.
>
> Our settings were designed to be as challenging as possible in terms of signal recovery: Large boxes are hard to inpaint, especially due to the inductive bias of the convolutional neural networks, as they are rather poor at incorporating nonlocal semantics; 92% random drop of pixels is challenging simply due to the extremely sparse amount of signal present in the measurement. Significant drop in LPIPS scores for state-of-the-art methods such as RePAINT and LaMa stems from such challenging settings. Furthermore, the poor performance of LaMa is also attributed to overfitting to the training settings, where the testing masks deviate from the masks that were used during training (especially random dropping). We would like to emphasize that our method is unsupervised and agnostic to any masking schemes, and is perfectly compatible with all the different masking schemes.
>
> Having said that, we agree that it would be beneficial to also include the experiments where we compare the models with LaMa masks. We show that our model is competitive with the state-of-the-art methods in such a regime. Please see the modified Table 1. We have also modified the appendix figures accordingly.
>
> We would also like to comment on the correctness of the experiments with the comparison methods. All methods including LaMa were tested with the official codebase, and with pre-trained checkpoints whenever possible. When pre-trained checkpoints were unavailable, we invested a considerable amount of time to tune the hyper-parameters to achieve the best result that we could get when training these models from scratch.
>
> For RePAINT, by the time when we were writing the manuscript, the official codebase was unavailable, and we had to resort to our own re-implementation (see code in our supplementary material), where we tried to abide closest to the algorithm introduced in the paper. A few days ago, the official code was released with several heuristics that were not explicitly mentioned in the paper, thereby inducing huge gains in performance. We re-ran all the experiments with RePAINT, with improved results. Note that **the proposed MCG method still outperforms RePAINT**, albeit using the official code.

---

> > ### Author Response · Authors · 2022-08-01
> > **Continuation of our response**
> >
> > **Q3. Misleading use of Bayes rule**
> >
> > A. The use of the Bayes rule for conditioning diffusion models on y similar to (12) in the revised text was indeed used in prior studies. That being said, in Eqs. (13) and (14), we try to induce yet additional constraint that will help the generative process on the manifold, and this is orthogonal to the y-conditioning that the prior works proposed with the Bayes rule. Hence, we would like to respectfully argue that our formulations for conditioning using the Bayes rule are not misleading.
> >
> > **References**
> >
> > [1] Suvorov, Roman, et al. "Resolution-robust large mask inpainting with fourier convolutions." WACV 2022.
> >
> > [2] Lugmayr, Andreas, et al. "Repaint: Inpainting using denoising diffusion probabilistic models." CVPR. 2022.
> >
> > [3] Dhariwal, Prafulla, and Alexander Nichol. "Diffusion models beat gans on image synthesis." NeurIPS 2021
> >
> > [4] Song, Yang, et al. "Score-based generative modeling through stochastic differential equations." ICLR 2021.
> >
> > [5] Chung, Hyungjin, Byeongsu Sim, and Jong Chul Ye. "Come-closer-diffuse-faster: Accelerating conditional diffusion models for inverse problems through stochastic contraction." CVPR 2022.

---

> > > ### Comment · Reviewer_vg3b · 2022-08-08
> > > **Thank you for your response**
> > >
> > > Thank you for your elucidating response. My main concern with the empirical results, which was that the comparisons were conducted unfairly, is greatly alleviated. I think it may behoove the authors to emphasize the fact that the proposed method excels at a more difficult task, and namely that the new inpainting conditions (e.g. Random and Extreme) are significantly harder than the previous baseline. This really helps explain the reduction in performance of competing methods (e.g., LaMa and RePAINT)
> > >
> > > On the theoretical side, I am happy the authors agree with my statement, and have tempered their theoretical claims. I am okay with the somewhat unrealistic assumptions of the theory in light of the empirical performance in the model (which I am now convinced by). Ultimately, I think it is very hard to be theoretically precise with any framework involving deep learning models, as their approximation properties are very difficult to define mathematically.
> > >
> > > I will raise my score, as I am now quite happy with the soundness of the work.

---

> ### Author Response · Authors · 2022-08-06
> **We are looking for your comments.**
>
> Dear Reviewer vg3b,
>
> As the discussion deadline is fast approaching, we would like to kindly remind the reviewer that we are looking forward to hearing your feedback. We have tried our best to address all your comments such as **1) comparison with RePAINT and LAMA with different masks, and 2) clarification of the main Theorem, and 3) use of the Bayes' rule.** Therefore, we would greatly appreciate it if our revision can properly address your concern.

---

### Official Review · Reviewer_mc79 · 2022-07-12

**Rating:** 8
**Confidence:** 4
**Soundness:** 4 excellent
**Presentation:** 4 excellent
**Contribution:** 4 excellent

**Summary:**

This paper deals with diffusion models for inverse problems. An important drawback of this approach is that to produce satisfactory results, it is necessary to iterate the forward and the reverse diffusions several times. Based on the manifold hypothesis, the authors show that this occurs because the score function only acts on the normal direction of the manifold, and therefore a drift occurs pushing the inference path out of the manifold. To solve this, besides the measurement constraint equation, they propose to include a constraint on the reverse diffusion equation given by the gradient of the $L^2$ residual of the inverse problem for the Bayes optimal denoising step from Noise2Noise (Kim and Ye, Neurips 2021). The authors prove that this constraint forces the diffusion to lie on the data manifold. These claims are supported by theoretical proofs and a clear geometric interpretation.


**Questions:**

None

**Limitations:**

Yes

**Strengths And Weaknesses:**

## Strengths

(+) The paper is very well written
(+) The idea of combining Noise2Score and diffusion models is novel, interesting, and completely sound.
(+) The proposed strategy is fully backed up by the theortical arguments and proofs provided by the authors.
(+) Thorough comparison with state-of-the-art diffusion model-based approaches and supervised learning-based baselines confirm the superiority of the proposed approach.

## Weaknesses

(-) It would be nice to show and discuss failure cases, or situations when the proposed approach does not outperform the others.

Minor comments:

- table X, figure Y, section Z, etc. --> Table X, Figure Y, Section Z, etc.
- Eq. 3: $x_t$ --> $x(t)$
- Fix punctuation at the end of Eqs. 6 and 9
- L71: $\mathbb{R}^n$ --> $\mathbb{R}^m$
- L76: utilize
- L77: rely
- Eqs. 13 and 14: for consistency, write them in the discrete setting
- L144: uses
- L154: recall the definition of $p_0$ (it was only defined in Section 2)
- SuppMat, L465-466: check the sentence
- SuppMat, Fig. 6: there are two lines in red that should be in green
 - SuppMat, L502: $\epsilon_\theta$ --> $z_\theta$
 - SuppMat,L507: (4) --> Table 4
 - SuppMat, L509: (1) --> Algorithm 1

---

> ### Author Response · Authors · 2022-08-01
> **Thank you for your encouraging review and feedbacks.**
>
> Thank you for your encouraging and thoughtful reviews. Below we would like to address your comments:
>
> **Q. Failure cases, and typos**
>
> A. For the diffusion model that is trained for ImageNet, we observe that when we perform inpainting by masking half of the region of the whole image, MCG often generates symmetric images that are not desirable. This behavior is not observed with our FFHQ experiments, and we conjecture the behavior is from the imperfect capacity of the diffusion model for learning ImageNet data set (as it is widely known to be a much harder dataset to learn). We have included this in our Limitations and broader impacts paragraph and included a section for the failure cases in the Appendix.
>
> For the minor comments, we would like to thank the reviewer for the careful reading. We have fixed the manuscript accordingly.

---

> ### Author Response · Authors · 2022-08-07
> **Follow up before the discussion is closed.**
>
> Thank you again for the review. As we are approaching the end of the discussion period, we would like to kindly ask the reviewer if our revision and response have acknowledged your questions. We would be happy to answer and discuss if you have further comments.

---

### Author Response · Authors · 2022-08-01
**Summary of updates, overall comment**

We would like to thank the reviewers for their constructive and thorough reviews. We are encouraged that the reviewers think that our paper is “novel, interesting, and completely sound” (mc79), “elegant, computationally simple, and suggests massive improvement in the signal recovery” (vg3b), “generally well written and the idea is well motivated” (qC4w), and “makes sense from a theoretical point of view, and easy to implement” (ze1d).

We have updated our draft to further clarify the theoretical aspects along with new experiments (and some corrections to the existing ones), extended the appendix to elaborate on the experiment settings, and discuss limitations. In the following, we summarize the major changes that were made to the manuscript.

**A. Strengthening and correcting experiments**

As LaMa wide masks are widely used as the baseline setting for measuring the performance of inpainting, we have included this setting in our inpainting experiment. IAGAN was included as a representative GAN-based inverse problem solver for inpainting. The results for RePAINT were updated with results obtained by running the official code from the authors.

**B. Ablation studies**

We conducted three ablation studies: 1) Clarify the contribution of two terms – MCG term, and the projection term. 2) Align all the diffusion model-based methods with the same NFE, and observe the performance as we vary it. 3) Inspect the behavior of MCG as we vary the step size $\alpha$.

**C. Strengthen introduction and related works**

Many of the reviewers introduced us to many relevant works that are worth discussing as related works. Hence, we have further extended our discussion ranging from plug and play methods to GAN-based ones and methods that leverage Tweedie’s formula.

**D. Temper theoretical claims**

To maximize the rigor in our statements, our theoretical analysis builds on rather strong assumptions: 1) Learned score function is globally optimal, and 2) Low dimensional manifold is locally linear. These may not be practical assumptions that holds in practice, and hence we now tone down our claims, making much clearer that the assumptions may not hold.

---

> ### Comment · Reviewer_qC4w · 2022-08-08
> **Updated manuscript and final comments.**
>
> I think that the modifications introduced by the authors help to clarify some of the issues raised in the reviews. I'm happy to update my score. I understand that this is not a finished work (more analysis might be needed) and there are still some open issues for discussion. But I believe that having this paper out will contribute to the community in a positive way (strengths >> weakness).

---

### Comment · Area_Chair_XGgP · 2022-08-03
**Discussion period**

Thanks to all reviewers and authors for their work on this submission.

As the discussion period starts, I want to make sure that reviewers have read the author's response.

This can be done either by communicating with authors, or in private conversation within the reviewing team.

---

### Meta-Review · Area_Chair_XGgP · 2022-08-20

**Recommendation:** Accept
**Confidence:** Certain

**Metareview:**

This paper proposes to incorporate manifold constraints in diffusion model (for inverse problems). The general consensus after rebuttal is that this submission is worthy of acceptance to NeurIPS. Please incorporate the remaining reviewers' feedback for the camera-ready version.

**Award:**

No

---

### Decision · Program_Chairs · 2022-09-14

Accept